# ON THE RELATION BETWEEN STATISTICAL LEARNING AND PERCEPTUAL DISTANCES

**Alexander Hepburn**
Engineering Mathematics
University of Bristol
`alex.hepburn@bristol.ac.uk`

**Valero Laparra**
Image Processing Lab
Universitat de Valencia
`valero.laparra@uv.es`

**Raul Santos-Rodriguez**
Engineering Mathematics
University of Bristol
`enrsr@bristol.ac.uk`

**Johannes Ballé**
Google Research
`jballe@google.com`

**Jesús Malo**
Image Processing Lab
Universitat de Valencia
`jesus.malo@uv.es`

## ABSTRACT

It has been demonstrated many times that the behavior of the human visual system is connected to the statistics of natural images. Since machine learning relies on the statistics of training data as well, the above connection has interesting implications when using perceptual distances (which mimic the behavior of the human visual system) as a loss function. In this paper, we aim to unravel the non-trivial relationships between the probability distribution of the data, perceptual distances, and unsupervised machine learning. To this end, we show that perceptual sensitivity is correlated with the probability of an image in its close neighborhood. We also explore the relation between distances induced by autoencoders and the probability distribution of the training data, as well as how these induced distances are correlated with human perception. Finally, we find perceptual distances do not always lead to noticeable gains in performance over Euclidean distance in common image processing tasks, except when data is scarce and the perceptual distance provides regularization. We propose this may be due to a *double-counting* effect of the image statistics, once in the perceptual distance and once in the training procedure.

## 1 INTRODUCTION

The relationship between the internal representations of supervised learning models and biological systems has previously been explored (Cadieu et al., 2014), and this connection can be explained by both systems being optimized to perform the same object recognition task. A much less studied area is comparing modern representations learned in an unsupervised manner to those of biological systems. As one of the most influential ideas in this area, the *efficient coding hypothesis* states that the internal representations of the brain have evolved to efficiently represent the stimuli (Attneave, 1954; Barlow, 1961) and has been validated against statistical models for images (Simoncelli & Olshausen, 2001; Malo & Laparra, 2010). Similarly, the explicit constraints of dimensionality reduction or compression present in many unsupervised representation learning models, mainly autoencoders (Ballé et al., 2018; 2016; Baldi, 2012), impose a type of parsimony on the representation. To understand properties of these representations of interest, here we consider distance measurements between the representations of pairs of stimuli. Such distance measurements can be thought of as *perceptual distances*, which are rooted in psychophysics: a good perceptual distance mimics the human rating of similarity between the two stimuli with high accuracy.

Traditionally, perceptual distances have been hand-designed models with few adjustable parameters, inspired by the physiology or observations in visual psychology, as the Multi-Scale Structural SIMilarity index (MS-SSIM) (Wang et al., 2003). More recently, it has become common to use an explicit image representation and 'induce' a distance from it. For instance, comparing the internal representations of models trained for image classification for pairs of stimuli has been used for

perceptual judgments and was shown to correlate well with human opinions (Zhang et al., 2018; Ding et al., 2020). This is also true for unsupervised representations that focus on learning features of natural scenes which are information efficient. For example, in the normalized Laplacian pyramid distance (NLPD) (Laparra et al., 2016), the representation is learned based on redundancy reduction in neighboring pixels. The Perceptual Information Metric (PIM) (Bhardwaj et al., 2020) uses a contrastive representation learning technique based on compression and slowness. With regards to training autoencoders, a particular type of model that can be used to unsupervisedly learn an explicit representation, here we examine three distinct types of induced distances: the reconstruction distance $D_r$, the inner distance $D_{in}$ and the self-distance $D_s$. These distances correspond to different representations learned by the autoencoder (see Fig. 1).

While the connection between the biological response and image probability has been examined before (Laughlin, 1981; Twer & MacLeod, 2001; Malo & Gutiérrez, 2006; Malo & Laparra, 2010; Laparra et al., 2012; Laparra & Malo, 2015; Hyvärinen et al., 2009), the relation between perceptual distances, unsupervised image representations, and the statistics of natural images has not been studied in depth. The current understanding is simply that distances induced by representations relevant for image classification or compression are useful for perceptual judgments. We show that the relation is deeper than that, linking it to image statistics.

Furthermore, we examine the unexpected effects of utilizing perceptual distances in the loss functions of autoencoders, which is a common approach in designing neural image compression methods (Ballé et al., 2018). One would expect that using a perceptual distance that is closely related to the internal image representation of the brain would give rise to a representation inside the model that is much more closely tied to humans, but that does not seem to be the case. This is surprising given the limited ability of Euclidean distances like Mean Squared Error (MSE) to reproduce human opinion (Girod, 1993; Wang & Bovik, 2009), compared to successful perceptual distances. We argue that this is because of a *double counting effect* where the distribution of natural images is taken into account twice in the training of the autoencoder; once in the training data and again in the perceptual distance, leading to an over-stressing of high density regions in the data. Conversely, where data is sparse or contains non-representative samples, this effect can result in regularization by discounting losses from outliers. Our specific contributions are:

**1**    We demonstrate that good perceptual distances, i.e. distances that are good at predicting human psychophysical responses, are also correlated with image likelihoods obtained using a recent probabilistic image model (PixelCNN++ (Salimans et al., 2017)). This underlines indirectly the idea that part of the biology is informed by efficient representation, as conjectured by Barlow.

**2**    The distances induced by autoencoders trained to minimize an Euclidean loss are correlated with the probability of the training data. Moreover, when autoencoders are trained using natural images, these induced distances are highly correlated with human perception.

**3**    Using a perceptual distance instead of a Euclidean loss in the optimization of autoencoders implies taking into account the data distribution *twice*: one in the perceptual distance and another through the empirical risk minimization procedure. We call this the *double counting effect*. This effect may explain the limited improvements obtained when using perceptual distances in some machine learning applications. We find that perceptual distances lead to models that over-stress high density regions. We emphasize this by showing that image autoencoders can be trained without image data (just using uniform random noise as input) if a proper perceptual distance is used.

## 2    ON THE RELATION BETWEEN PERCEPTUAL DISTANCE AND IMAGE PROBABILITY

The broadest formulation of the classical *efficient coding hypothesis* in neuroscience (Barlow, 1961) states that the organization of the sensory system can be explained by the regularities of the natural signals (Barlow, 2001). This hypothesis has inspired statistical derivations of the nonlinear response of biological vision based on equalization: maximization of channel capacity (Laughlin, 1981) and error minimization under noise (Twer & MacLeod, 2001) are one-dimensional equalization techniques that directly relate signal probability, $p(\boldsymbol{x})$, with the response of the system to the visual input, $\boldsymbol{y} = S(\boldsymbol{x})$, where $S$ is the given system. These univariate equalization techniques have been generalized to multivariate scenarios to explain the nonlinearities of the biological responses to

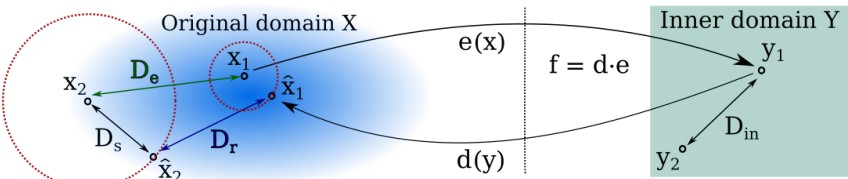

Figure 1: **Distances in autoencoders**. $D_e$ is the euclidean distance (green), $D_r$ is the *reconstruction distance* (blue), $D_s$ is the *self-distance*, $D_{in}$ is the *inner distance*. Red circles illustrate the *allowed error* for the points $\mathbf{x}_1$ and $\mathbf{x}_2$, i.e. the reconstructed points ($\hat{\mathbf{x}}_1$ and $\hat{\mathbf{x}}_2$) should live inside the circle. The sizes are inversely proportional to the density (blue background). The autoencoder function, $f$, is the composition of the encoder function, $e$, and the decoder function, $d$.

spatial texture (Malo & Gutiérrez, 2006), color (Laparra et al., 2012), and motion (Laparra & Malo, 2015). Interestingly, biological networks optimized to reproduce subjective image distortion display statistical properties despite using no statistical information in fitting their parameters. Divisive normalization obtained from perceptual distances substantially reduces the redundancy in natural images (Malo & Laparra, 2010), and captures a large fraction of the available information in the input (Malo, 2020). A similar factorization is obtained in Wilson-Cowan networks fitted to reproduce subjective distortions (Gomez-Villa et al., 2020a). Here the perceptual distance between an image $\mathbf{x}_1$ and its distorted version, $\mathbf{x}_2$, will be referred to as $D_p(\mathbf{x}_1, \mathbf{x}_2)$ and is computed in the response representation $\mathbf{y}$. For small distortions (i.e. $\mathbf{x}_1$ close to $\mathbf{x}_2$), equalization models imply that there is an explicit relation between the perceptual distance, $D_p(\mathbf{x}_1, \mathbf{x}_2)$, and the probability of the original image according to the distribution of natural images, $p(\mathbf{x}_1)$ (see Appendix A):

$$\frac{D_p(\mathbf{x}_1, \mathbf{x}_2)}{||\mathbf{x}_2 - \mathbf{x}_1||_2} = \frac{D_p(\mathbf{x}_1, \mathbf{x}_2)}{\sqrt{m}\,\text{RMSE}} \approx p(\mathbf{x})^\gamma \tag{1}$$

where $\gamma > 0$, and the Euclidean length $||\mathbf{x}_2 - \mathbf{x}_1||_2$ is just the Root Mean Squared Error (RMSE) scaled by the dimensionality of the data $m$. As we are interested in correlations, ignoring the scaling term, this relation leads to the first observation that we check experimentally in this work:

**Observation 1** *The perceptual quantity $\frac{D_p(\mathbf{x}_1, \mathbf{x}_2)}{RMSE(\mathbf{x}_1, \mathbf{x}_2)}$ is correlated with $p(\mathbf{x}_1)$ for small distortions (i.e. when $||\mathbf{x}_1 - \mathbf{x}_2||_2 < \delta$ for small $\delta$).*

This quantity is related to the sensitivity of the perceptual distance to displacements in the signal domain, $\frac{\partial D_p}{\partial \boldsymbol{x}}$. In fact, the (n-dimensional) gradient reduces to $\frac{D_p}{\text{RMSE}}$ in the 1-d case (Appendix A.1). The observation states that the perceptual distance is more sensitive in high probability regions.

This observation is experimentally illustrated in Appendix A.2, that shows a strong connection between the sensitivity of perceptual distances (either NLPD or MS-SSIM) and the distribution of natural images, as opposed to the sensitivity of Euclidean RMSE, which is not related to the data.

## 3 ON THE RELATION BETWEEN DISTANCES INDUCED BY AUTOENCODERS AND PROBABILITY

In section 2 we showed the relation between perceptual distances and the distribution of natural images, and here we connect the distance induced by a statistical model and the probability of the data used to train it. We first discuss our observations, followed by describing the experimental setup.

Statistical learning is based on risk minimization which connects the model, the loss to minimize and the distribution of the data. As a result, the trained model captures the statistics of the data. In this section we elaborate on this connection by computing Euclidean distances in the representations induced by the learned model. Here we focus on autoencoders, although the considerations apply to other statistical learning settings. We will consider the three different options to define the induced distances by an autoencoder shown in Fig. 1. We will refer to them as *self-reconstruction* distance $D_s = ||\mathbf{x} - \hat{\mathbf{x}}||_2$, *reconstruction* distance $D_r = ||\hat{\mathbf{x}}_1 - \hat{\mathbf{x}}_2||_2$, and *inner* distance $D_{in} = ||\mathbf{y}_1 - \mathbf{y}_2||_2$. The autoencoder model will be denoted as $f$, which is consists of an encoder $e(\cdot)$ and a decoder $d(\cdot)$, $f = d \circ e$. The reconstructed data is, $\hat{\mathbf{x}} = f(\mathbf{x})$, and $\mathbf{y}$ is the data in the inner domain, $\mathbf{y} = e(\mathbf{x})$. We will explicitly show how all these three distances depend in different ways on the probability of the

input data.

Given samples $\mathbf{x}$ from a distribution $p(\mathbf{x})$ a generic autoencoder minimizes this risk:

$$R = \mathbb{E}_{\mathbf{x}}[\mathcal{L}(f(\mathbf{x}), \mathbf{x})] = \int \mathcal{L}(f(\mathbf{x}), \mathbf{x}) dP(\mathbf{x}) = \int p(\mathbf{x}) \cdot \mathcal{L}(\hat{\mathbf{x}}, \mathbf{x}) d\mathbf{x} \qquad (2)$$

where $\mathbb{E}_{\mathbf{x}}[\cdot]$ stands for expectation over variable $\mathbf{x}$ $\mathcal{L}$ is a loss between the input $\mathbf{x}$ and the reconstruction $f(\mathbf{x}) = \hat{\mathbf{x}}$. Since $p(\mathbf{x})$ is unknown and often intractable, the risk is approximated using the empirical risk minimization principle (Devroye et al., 1996; Vapnik, 1999): $R_{\text{emp}} = \frac{1}{n} \sum_{i=1}^{n} \mathcal{L}(f(\mathbf{x}_i), \mathbf{x}_i)$, where the $\mathbf{x}_i$ are samples from the training dataset of size $n$. Although it is well known, it is important for the reasoning of this work to stress that, the function $f(\mathbf{x})$ that minimizes the empirical risk $R_{\text{emp}}$ is not the same function that would minimize the loss function $\mathcal{L}$ uniformly over all space. For example, if we choose $\mathcal{L} = ||\mathbf{x_i} - f(\mathbf{x_i})||_2$, minimizing $R_{\text{emp}}$ is not minimizing the RMSE, but the RMSE weighted by the probability of samples, Eq. 2. Therefore, the distance an autoencoder induces will be different to the loss function used for training. Once the model, $f(\mathbf{x})$, is trained, it inherits some properties of the data distribution, $p(\mathbf{x})$. This implies that the model's behavior depends on the probability of the region of the input space where it is applied.

In what follows, we make some observations on the relation between the distances $D_s$, $D_r$, and $D_{in}$ and the distribution $p(\mathbf{x})$. We assume that $\mathcal{L}$ is the Euclidean distance (or RMSE), but similar arguments qualitatively apply to distances monotonically related to RMSE.

**Observation 2** *The self-reconstruction distance in an autoencoder is correlated to the inverse of the probability:* $D_s = ||\mathbf{x} - \hat{\mathbf{x}}||_2 \propto \frac{1}{p(\mathbf{x})}$.

The difference between the original and the reconstructed data point in denoising and contractive autoencoders has been related with score matching: $\mathbf{x} - \hat{\mathbf{x}} = \frac{\partial log(p(\mathbf{x}))}{\partial \mathbf{x}}$ (Vincent, 2011; Alain & Bengio, 2014). This expression can be formulated using the distribution of the noisy data instead (Miyasawa, 1961; Raphan & Simoncelli, 2011). Here we argue that for a family of distributions the modulus of the difference ($||\mathbf{x} - \hat{\mathbf{x}}||_2$) can be related with the distribution itself (Obs. 2). This is true for Gaussian distribution, and a good proxy in general for unimodal distributions as the ones proposed to describe the distribution of natural images (Portilla et al., 2003; Hyvärinen et al., 2009; Lyu & Simoncelli, 2009; Malo & Laparra, 2010; Lyu, 2011; van den Oord & Schrauwen, 2014; Ballé et al., 2016) (see details in Appendix B).

Our intuition comes from Eq. 2 that enforces small reconstruction errors in high probability regions, i.e. when $p(\mathbf{x})$ is high, $||\mathbf{x} - f(\mathbf{x})||_2$ should be low. This implies a dependency for the *allowed error* (variance of the error) on the probability: $Var_x(||\mathbf{x} - f(\mathbf{x})||_2) \propto \frac{1}{p(\mathbf{x})}$, where $Var_x$ is computed around the point $\mathbf{x}$. This is the idea behind the use of autoencoders for anomaly detection. An autoencoder trained for data coming from a particular distribution $p(\mathbf{x})$, obtains high self-reconstruction error when it is evaluated on data from a different distribution.

Analogous to the distance used in Observation 1, we argue that the quantity $\frac{D_r}{\text{RMSE}}$ is correlated with the probability. The rationale is that if $f(.)$ was trained to minimize the average distortion introduced in the reconstruction, then $D_r$ has to be more sensitive in high density regions.

**Observation 3** *The sensitivity of the reconstruction distance induced by an autoencoder in* $\mathbf{x}_1$, $\frac{D_r}{RMSE} = \frac{||f(\mathbf{x}_1) - f(\mathbf{x}_2)||_2}{||\mathbf{x}_1 - \mathbf{x}_2||_2}$, *is correlated with* $p(\mathbf{x}_1)$ *when* $||\mathbf{x}_1 - \mathbf{x}_2||_2 < \delta$ *for small* $\delta$.

While the autoencoder aims to reconstruct the original data, there are some restrictions which make the learned transformation different from the identity. This transformation is going to put more resources in regions where training data is more likely. This enforces a relation between the sensitivity (derivative) of the model and the probability distribution, i.e. $|\frac{\partial f}{\partial \mathbf{x}}(\mathbf{x})| \not\perp p(\mathbf{x})$ (see an example in Appendix C). Our hypothesis is that this relation should be positively (possibly non-linearly) correlated. In particular we will be reasoning in the direction of using $p(\mathbf{x}_1)||\mathbf{x}_1 - \mathbf{x}_2||_2$ as a proxy for the induced reconstruction distance $D_r$ when $\mathbf{x}_1$ and $\mathbf{x}_2$ are close.

While the distance in the inner domain, $D_{in} = ||\mathbf{y}_1 - \mathbf{y}_2||_2$, is highly used (Ding et al., 2020; Zhang et al., 2018; Gatys et al., 2016), it is the most difficult to interpret. The main problem is to decide which representation inside the model to use. For autoencoders, the inner representation is usually selected as the layer with the least dimensions, although it is arbitrary what is defined as the inner representation. In compression autoencoders one has two suitable candidate representations, before or after the quantisation step. Both have different interesting properties, however in either cases the

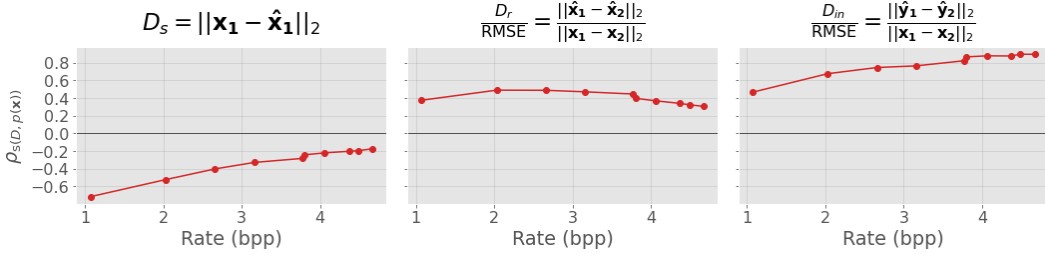

**Figure 2: Illustrating Observations 2, 3, and 4.** Spearman correlations $\rho_S(D, p(\mathbf{x_1}))$ between different distances and the probability of point $\mathbf{x_1}$ are shown. Each point corresponds to the correlation for one autoencoder trained for a particular rate regime.

sensitivity conjecture also makes qualitative sense for $D_{in}$.

**Observation 4** *The sensitivity of the distance induced by a compression autoencoder in the inner domain at point $\mathbf{x}_1$, $\frac{D_{in}}{RMSE} = \frac{||e(\mathbf{x}_1) - e(\mathbf{x}_2)||_2}{||\mathbf{x}_1 - \mathbf{x}_2||_2}$, correlates with $p(\mathbf{x}_1)$ when $||\mathbf{x}_1 - \mathbf{x}_2||_2 < \delta$ for small $\delta$.*

This observation has a strong connection with density estimation, independent component analysis and data equalization. The density target under a transformation is $p(\mathbf{x}) = p(e(\mathbf{x}))|\frac{\partial e}{\partial \mathbf{x}}(\mathbf{x})|$. If one wants to describe p($\mathbf{x}$), a suitable target distribution for the transform domain is the uniform distribution. Therefore, if one assumes $p(e(\mathbf{x}))$ constant, the determinant of the Jacobian of the transformation is equal to the probability distribution as assumed in the equalization model (Observation 1), and similarly, one would have $\frac{||e(\mathbf{x}) - e(\mathbf{x} + \Delta_x)||_2}{||\mathbf{x} - \mathbf{x} + \Delta_x||_2} \approx p(\mathbf{x})$. A similar result can be obtained modeling the noise in the inner domain (Berardino et al., 2017). Note this observation has parallelisms with the procedure that the human visual system has to do in order to process the natural images as stated in Sec. 2.

**Experiment.** Here we explore the relation of the previously defined distances with the probability distribution of the training data using a compression autoencoder in a toy 2D example. Compression using autoencoders is presented as follows (Ballé et al., 2018; 2016); an input $\mathbf{x}$ is transformed using the encoder, or analysis transform, $e(\mathbf{x})$ to an encoding $\mathbf{y}$. Scalar quantisation is then applied to $\mathbf{y}$ to create a vector of integer indices $\mathbf{q}$ and a reconstruction $\hat{\mathbf{y}}$. The quantisation is approximated during training with additive uniform noise $\tilde{\mathbf{y}} = \mathbf{y} + \Delta \mathbf{y}$ to get a continuous gradient. It is then transformed back into the image domain using the decoder $d(\hat{\mathbf{y}})$ to create the reconstructed image $\hat{\mathbf{x}}$.

Given $\mathbf{x}$, the aim is to find a quantised representation $\mathbf{q}$ that minimizes the approximated entropy, or rate $\mathcal{R} = H[p(\tilde{y})]$, estimated by a density model $p$, whilst also minimizing the distortion loss, $\mathcal{D}$, usually weighted by a Lagrange multiplier in order to achieve different compression rates. Thus the loss function $\mathcal{L} = \mathcal{R} + \lambda \mathcal{D}$ becomes

$$\mathcal{L} = \mathbb{E}_{\mathbf{x}}[-\log_2 p(e(\mathbf{x}) + \Delta \mathbf{y})] + \lambda \mathcal{D}(\mathbf{x}, d(e(\mathbf{x}) + \Delta \mathbf{y})). \tag{3}$$

In order to have explicit access to the data distribution, a 2D Student-t distribution is used with one dimension having high variance and the other low variance. The Student-t distribution is a reasonable approximation for the distribution of natural images (van den Oord & Schrauwen, 2014). A 2 layer MLP is used for $e$ and $d$. For full experimental setup see Appendix C. The induced distances are calculated for a set of samples taken from the Student-t distribution. Fig. 2 shows that these induced distances are correlated with the probability $p(\mathbf{x_1})$. The three observations hold for intermediate regimes, low rate gets bad reconstruction while high rate gets almost no compression. Observation 2 ($D_s$) holds at low and medium rates where the Spearman correlation is -0.68 to -0.4, but starts to fail at 4.67bpp (bits-per-pixel) where correlation approaches -0.18. The same occurs for Observation 3 ($D_r$) where at high entropy the correlation decreases slightly from 0.49 at 2.03bpp to 0.31 at 4.67bpp. Observation 4 ($D_{in}$) holds over all rates but especially at high rates where the largest correlation of 0.90 is found. These results indicate that the learned representations inside an autoencoder correlate with the training distribution. Although we test at a wide range of rates, if the entropy is very restricted, $f = d \cdot e$ has no capacity and in the limit may even be a constant function. However, this is a very extreme case and, in practice, an autoencoder like this would never be used.

## 4 ON THE RELATION BETWEEN DISTANCES INDUCED BY AUTOENCODERS AND PERCEPTION

As stated above, machine learning models trained on natural images for classification or compression can lead to successful perceptual distances (Zhang et al., 2018; Ding et al., 2020; Bhardwaj et al., 2020). Similarly, human-like Contrast Sensitivity Functions arise in autoencoders for image enhancement (Gomez-Villa et al., 2020b; Li et al., 2021). Reproducing CSFs is key in perceptual distances (Mannos & Sakrison, 1974; Watson, 1993). The latent space of a machine learning model can also be used for smooth interpolation between images (Connor & Rozell, 2020; Berthelot et al., 2018) whilst the statistics of activations in different layers can be used to interpolate between textures (Vacher et al., 2020).

The emergence of perceptually relevant distances (or key features to compute perceptual distances) in image classifiers and autoencoders trained to optimize Euclidean distortions over natural images should not be a surprise given the correlations found in Sections 2 and 3: both the perceptual distances and the autoencoder-induced distances are correlated with the probability. Therefore, these two distances should be correlated as well, and more importantly, with the actual opinion of humans. We indeed find this to be the case:

**Observation 5** *Both the reconstruction distance induced by an autoencoder, $D_r(\mathbf{x}_1, \mathbf{x}_2) = ||f(\mathbf{x}_1) - f(\mathbf{x}_2)||_2$, and the inner distance, $D_{in}(\mathbf{x}_1, \mathbf{x}_2) = ||e(\mathbf{x}_1) - e(\mathbf{x}_2)||_2$ are correlated with the subjective opinion given by humans.*

**Experiment.** The TID 2013 dataset (Ponomarenko et al., 2013) contains 25 reference images, $\mathbf{x}_1$, and 3000 distorted images, $\mathbf{x}_2$, with 24 types and 5 levels of severity of distortion. Also provided is the mean opinion score (MOS) for each distorted image. This psychophysical MOS represents the subjective distortion rating given by humans which is the ground truth for perceptual distances – precisely what human vision models try to explain.

For all experiments, networks to compute $D_r$ and $D_{in}$ are pretrained models from Ballé et al. (2016)[1]. See the architecture details in Appendix D.

As in Section 3, the autoencoder induces distances in the reconstruction and inner domain. Fig. 3 reports the Spearman correlations between the induced distance and MOS, $\rho(D, \text{MOS})$, for the TID 2013 database. As a reference, we include the correlation between the MOS and RMSE $= ||\mathbf{x}_1 - \mathbf{x}_2||_2$. Results in Fig. 3 confirm Observation 5: correlations are bigger for both induced distances than for RMSE, particularly for $D_{in}$ (with a maximum of 0.77). The correlations for models trained with MSE or MS-SSIM are very similar in medium to high rates where the difference in low rates can be explained by the double counting effect described in Sec. 5.1. Correlations are bigger in the inner representation than the reconstructed representation, as in the 2D example in Fig. 2. For both induced distances, the maximum correlation for the reconstructed domain occurs at the highest bit-rate (more natural images). Interestingly, for networks trained using the perceptual distance MS-SSIM the correlations remain similar. It is common practice to use $D_{in}$ distance as a more informative distance between two data points than $D_r$, as with LPIPS (Zhang et al., 2018), DISTS (Ding et al., 2020) and PIM (Bhardwaj et al., 2020), where the correlation these methods achieve with human opinion scores is explained by the correlation in the inner domain.

It should be noted that specific examples of a reference and distorted image with the same internal representation, $e(\mathbf{x}_1) = e(\mathbf{x}_2)$, are similar to metamers in humans - physically different objects that elicit the same internal representation and thus have 0 perceptual distance (Freeman & Simoncelli, 2011). Whilst we could find images with maximum RMSE in the image domain but with the embedded distance being 0, the existence of these data points not included in TID does not reduce the validity of the experiment.

## 5 OPTIMIZING MACHINE LEARNING MODELS WITH PERCEPTUAL DISTANCES

In Sec. 2 it was established that the sensitivity of perceptual distances at a point is correlated with the probability of that point and in Sec. 3 we showed that distances induced by autoencoder

---

[1]Code taken from `https://github.com/tensorflow/compression`, under Apache License 2.0.

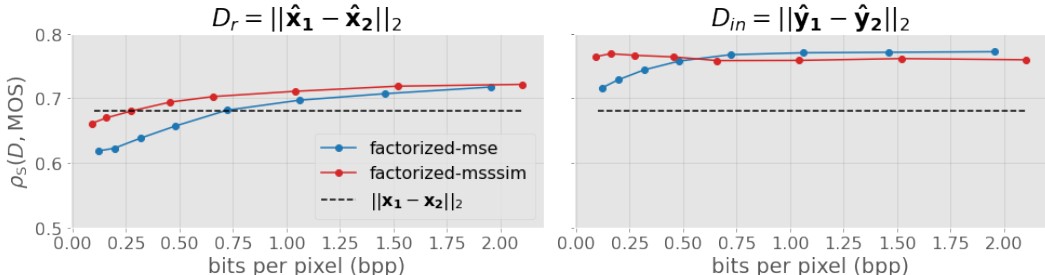

Figure 3: **Illustrating Observation 5**. Spearman correlations $\rho_S$ between induced distances ($D_r$ or $D_{in}$) and mean opinion score (MOS) for images from TID 2013 dataset (Ponomarenko et al., 2013). Pretrained compressive autoencoders at different bitrates were used. *factorized-mse* denotes networks trained using MSE $\mathcal{D} = ||\mathbf{x}_1 - \mathbf{x}_2||_2^2$ in Eq. 3, and *factorized-msssim* networks use $\mathcal{D} = 1 - \text{MS-SSIM}(\mathbf{x}_1 - \mathbf{x}_2)$ in Eq. 3.

representations also inherit a correlation with the probability of the point (which makes them acquire perceptual properties as seen in Sec. 4). This raises the question whether perceptual distances can be used *in place of* the data distribution. Empirically, we find this to be the case:

**Observation 6** *Optimizing using a perceptual distance $D_p$ as loss function has a similar effect as optimizing using the RMSE weighted by the probability,* $\arg\min \sum_{i=0}^{n} D_p(\mathbf{x}_i, \hat{\mathbf{x}}_i) \approx \arg\min \sum_{i=0}^{n} p(\mathbf{x}_i) \cdot ||\mathbf{x}_i, \hat{\mathbf{x}}_i||_2$.

Using a perceptual distance in an empirical risk minimization context (Eq. 2) appears to have a *double counting effect* of the distribution of natural images; once in the sensitivity of the perceptual distance and a second time when optimizing over the set of images.

In Sec. 5.1 we show that using a perceptual distance has a similar effect to using MSE weighted by $p(\mathbf{x})$. Excessive consideration of $p(\mathbf{x})$ may explain why using perceptual distances instead of MSE has little gain in some machine learning tasks (Hepburn et al., 2020; Ballé et al., 2016) despite the large difference in replicating perceptual judgments.

In Sec. 5.2 we present an extreme special case of this setting, where one has no samples from the distribution to minimize over. Instead, the model has direct access to the perceptual distance, which acts as a proxy of $p(\mathbf{x})$. This effect can be also seen in Fig. 3 where the difference of using or not using a perceptual distance can only be observed at low rates (poor quality / non-natural images): when dealing with poor quality data, extra regularization is helpful. This regularization effect is consistent with what was found when using perceptual distances in denoising (Portilla, 2006). Finally, in Appendix E.1 we show the effect when using a batch size of 1 in training.

### 5.1 PERCEPTUAL DISTANCES AND PROBABILITY HAVE A SIMILAR EFFECT IN RISK MINIMIZATION

Here we examine this effect in compression autoencoders, using either RMSE, RMSE weighted by the probability or a perceptual distance as a loss function. State of the art density estimators like PixelCNN++ can be used to estimate $p(\mathbf{x})$ only on small patches of images. On the other hand, perceptual distances are designed to estimate distances with large images. As such, we choose to compare the effect in two different scenarios: the 2D example used in Sec. 3 and the image compression autoencoders used in Sec. 4.

The rate/distortion is used as a proxy for the general performance of a network. The relative performance is obtained by dividing the performance of the network optimized for the RMSE weighted by the probability (or the perceptual distance) by the performance of the network using RMSE. A relative performance below $1.0$ means a performance gain using the probability (or perceptual distance). We evaluate the performance with data sampled from a line through the distribution from high to low probability. Fig. 4 shows the relative performance for both the 2D example and sampled images. For the 2D case, sampling through the distribution is trivial, $(x, y) \in ([0, 35], 0)$. For images, we modify image contrast of the Kodak dataset (Kodak, 1993) in order to create more and less likely samples under the image distribution (low contrast images are generally more likely (Frazor & Geisler, 2006)). $\alpha$ dictates where in the contrast spectrum the images lie,

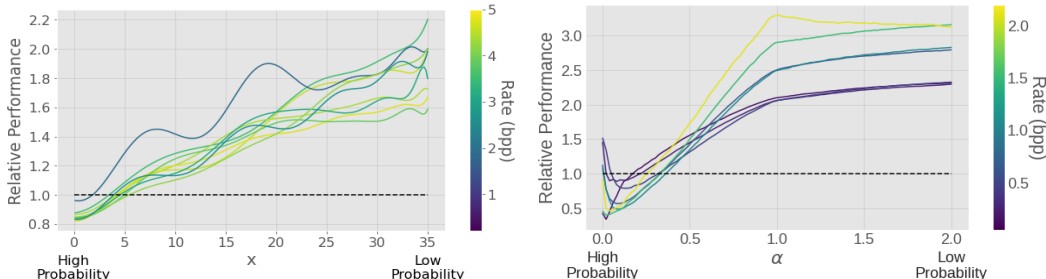

Figure 4: **Illustrating observation 6**. Figure shows the relative performance of the models for samples across the support of the respective data distributions. Both training with the probability (left) and the perceptual distance (right) cause the model to allocate more resources to high probability regions. Left: models trained with $\mathcal{D} = p(\mathbf{x}) \cdot ||\mathbf{x}_1 - \mathbf{x}_2||_2^2$ in Eq. 3 divided by performance of models trained with $\mathcal{D} = ||\mathbf{x}_1 - \mathbf{x}_2||_2^2$ on the 2D Student-t and evaluated using samples along $x$-axis (see Appendix C). Right: models trained for image compression with $\mathcal{D} = 1 - \text{MS-SSIM}(\mathbf{x}_1, \mathbf{x}_2)$ in Eq. 3 divided by performance of models trained with $\mathcal{D} = ||\mathbf{x}_1 - \mathbf{x}_2||_2^2$.

where $\alpha = 0.0$ represents very low contrast (solid gray), $\alpha = 1.0$ is the original image and $\alpha = 2.0$ illustrates high contrast. Details on how the performance is computed and how the samples from different probability regions are taken are in Appendix D.

We observe a performance gain in data points that are more probable and a performance loss away from the center of the distribution. This behavior is similar in both the Student-t example when multiplying the distortion by the probability, and when using a perceptual distance as the distortion term with images. In the 2D example (a) the maximum average performance gain across rates at the center of the distribution is 0.86, whereas with images (b) the maximum average performance gain is 0.64 at $\alpha = 0.08$. This leads us to the conclusion that multiplying by the probability and using a perceptual distance have a similar effect on the learning process.

## 5.2 TRAINING WITHOUT DATA

While in Appendix E.1 we show the effect of using a batch size of 1 during training, here we focus on the most extreme example. We show how explicit access to the samples from the data distribution might not be needed. If given the probability distribution, one can weight samples drawn from a uniform distribution with the same dimensionality and evaluate the likelihood of these points belonging to the original distribution. With the 2D Student-t example, using a uniform distribution and minimizing a loss weighted by the likelihood the samples belong to the Student-t, where the optimization would be $\min_{\mathbf{x} \sim \mathcal{U}} \mathcal{L} = \min_{\mathbf{x} \sim \mathcal{U}} p_\tau(\mathbf{x})^{0.1} \cdot (\mathcal{R} + \lambda \mathcal{D})$, where $p_\tau(\mathbf{x})$ denotes the probability of point $\mathbf{x}$ belonging to the 2D Student-t distribution (see Fig. 13b in Appendix C). In the 2D example this is trivial, however this is not the case for natural images. In Sec. 2, it can be seen that perceptual distances encode some information about the distribution natural images and, as such, can be used in a similar fashion to $p_\tau(\mathbf{x})$ when training on uniform distributed data. The perceptual distance can be minimized over uniform noise and should result in realistic looking images.

We will use the previous setting of compressive autoencoders but for ease of training, we approximate the rate-distortion equation in Eq. 3. Instead of optimizing the rate, we set an upper bound on the entropy of the encoding by rounding the points to a set of known code vectors $L$ and purely optimize for the distortion (Ding et al., 2021; Agustsson et al., 2019). We use two scenarios, $L = \{-1, 1\}$ leading to an entropy of 0.25 bpp and $L = \{-2, -1, 0, 1, 2\}$ 0.5 bpp. We refer the reader to Appendix E for more details.

Fig. 5 shows the result of minimizing MSE, NLPD and MS-SSIM over uniform data. Clearly, the networks trained with perceptual distances NLPD and MS-SSIM perform significantly better. The mean PSNR at 0.25 bpp over the Kodak dataset for the network trained using MSE is 13.17, for MS-SSIM is 18.92 and for NLPD is 21.21. In Appendix D we report results for each network evaluated with respect to PSNR, MS-SSIM and NLPD at 0.25 bpp and at 0.50 bpp. Using these perceptual distances significantly reduces our need for data. Although we have not explored how the amount of data scales affects the reconstructions, the extreme of having no data from the distribution of natural images but still obtaining reasonable reconstructions is a clear indication of the relationship

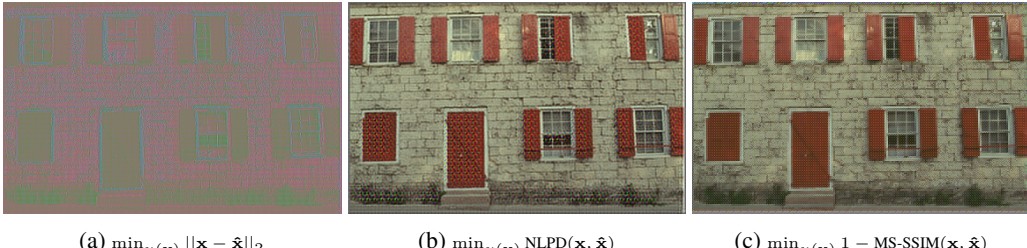

(a) $\min_{u(\mathbf{x})} ||\mathbf{x} - \hat{\mathbf{x}}||_2$    (b) $\min_{u(\mathbf{x})} \text{NLPD}(\mathbf{x}, \hat{\mathbf{x}})$    (c) $\min_{u(\mathbf{x})} 1 - \text{MS-SSIM}(\mathbf{x}, \hat{\mathbf{x}})$

Figure 5: **Illustrating observation 6**. The perceptual distance provides regularization when one has no samples from the training distribution. Decoded image (compressed at 0.25 bpp) encoded with networks trained using data from a uniform distribution and Euclidean vs. perceptual losses.

between perceptual distances, statistical learning and probability.

## 6 FINAL REMARKS

**Discussion**    Statistical learning and perceptual distances are related through their dependence on the probability distribution of the input data. While for statistical learning this dependence is direct and by construction, for perceptual distances the dependence comes from the evolution of human vision to efficiently process natural images. We are not the first (at all) arguing that human perception has to be related with the statistics of the natural images. However, the results presented here explicitly compare reasonable perceptual distances with accurate models of the PDF of natural images for the first time.

Taking inspiration from equalization-based perception models, we show that the sensitivity of perceptual distances is related to the distribution of natural images (Sec. 2). This is further shown experimentally in Appendix A. We also show that the distances induced by an autoencoder are correlated with the distribution used to train it. In particular, similarly to perceptual distances, the sensitivity of some of these autoencoder-induced distances are positively correlated with the PDF of images (Sec. 3, Fig. 2). In image compression, without imposing perceptual properties at all, the autoencoder-induced distances present a high correlation with human perception – a maximum Spearman correlation of 0.77 (Sec. 4, Fig. 3). The goal in compression is highly related with the *efficient coding hypothesis* proposed by Barlow to explain the organization of human vision. Note that a widely accepted perceptual distance like SSIM has a substantially lower Spearman correlation with psychophysical MOS in the same experiment ($\rho(SSIM, MOS) = 0.63$ (Ponomarenko et al., 2013) while $\max(\rho(D_{in}, MOS)) = 0.77$).

Our observations suggest that there is a double counting effect of the probability distribution when optimizing models empirically and using perceptual distances. This can be explained by the use of the data in the expectation, but also implicitly in the perceptual distance. Consequently, perceptual distances can cause the model to over-allocate resources to high likelihood regions (Sec. 5.1, Fig. 4). On the positive side, they can act as helpful regularizers in the case of limited or no data (Sec. 5.2, Fig. 5). By minimizing a perceptual distance over uniform noise, we show it is possible to successfully reconstruct images with a compressive autoencoder at low entropy. This effect has implications on the design of generative models, showing that perceptual distances can help when training with limited datasets where the the statistics of the dataset at hand might not match those of natural images.

**Conclusion**    We present an empirical exploration of the three-way relationship between (a) image representations (either learned as autoencoders, or natural, as perception models), (b) the distribution of natural images, and (c) actual human perceptual distances. We describe the experimental setup we used to examine this relationship and summarize our findings in six concise observations. Naturally, our experiments have limitations, such as regarding choices of perceptual and density models (MS-SSIM, PixelCNN++, etc.). However, the presented observations represent a step towards a better understanding of how both machine learning and biological perception is informed by the distribution of natural images.

ACKNOWLEDGMENTS

This work was partially funded by EPSRC grant EP/N509619/1, UKRI Turing AI Fellowship EP/V024817/1, Spanish Ministry of Economy and Competitiveness and the European Regional Development Fund under grants PID2020-118071GB-I00 and DPI2017-89867-C2-2-R, and the regional grant GV/2021/074.

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

# A ON THE RELATION BETWEEN PERCEPTUAL DISTANCE AND IMAGE PROBABILITY

## A.1 DERIVING EQUATION 1

Classical literature in psychophysics (Mannos & Sakrison, 1974; Watson, 1993; Laparra et al., 2010) assumes that the perceptual distance (subjective distance) between two images is given by the Euclidean difference between the inner representations of the original and the distorted images, $S(\boldsymbol{x}_1)$ and $S(\boldsymbol{x}_2)$, where $S(\cdot)$ is the function that represents the response of the biological visual system. Given the non-trivial nature of $S$, the perceptual distance in the input domain is non-Euclidean, and described by a non-diagonal (and eventually image dependent) perceptual metric matrix, $W$:

$$D_p(\boldsymbol{x}_1, \boldsymbol{x}_2) = ||\Delta\boldsymbol{y}||_2 \approx \sqrt{(\mathbf{x}_2 - \mathbf{x}_1)^\top \cdot W \cdot (\mathbf{x}_2 - \mathbf{x}_1)} \tag{4}$$

where, following (Epifanio et al., 2003; Malo et al., 2006), $W = \frac{\partial S}{\partial \boldsymbol{x}}^\top \cdot \frac{\partial S}{\partial \boldsymbol{x}}$. Assuming a 1-dimensional scenario for simplicity:

$$D_p(\boldsymbol{x}_1, \boldsymbol{x}_2) \approx \frac{\partial S}{\partial \boldsymbol{x}} \ ||\mathbf{x}_2 - \mathbf{x}_1||_2 \tag{5}$$

Now, following the perceptual equalization literature (Laughlin, 1981; Twer & MacLeod, 2001; Malo & Gutiérrez, 2006; Laparra et al., 2012; Laparra & Malo, 2015), the determinant of the Jacobian of the response is related to a monotonic function of the probability, $|\frac{\partial S}{\partial \boldsymbol{x}}| = p(\boldsymbol{x})^\gamma$, and then, we get Eq. 1:

$$\frac{D_p(\boldsymbol{x}_1, \boldsymbol{x}_2)}{||\mathbf{x}_2 - \mathbf{x}_1||_2} = \frac{D_p(\boldsymbol{x}_1, \boldsymbol{x}_2)}{\sqrt{m}\ \text{RMSE}} \approx p(\boldsymbol{x})^\gamma \tag{6}$$

where $\gamma > 0$ (typically $\gamma = 1$ for uniformization and $\gamma = \frac{1}{3}$ for error minimization) and $m$ is the dimensionality of the data.

The fact that Eq. 1 is a descriptor of the sensitivity of the distance is easy to see in the 1-d case. Note that the n-dimensional gradient vector is (Martinez et al., 2018): $\frac{\partial D_p}{\partial \boldsymbol{x}} = \frac{1}{D_p}\Delta S^\top \cdot \frac{\partial S}{\partial \boldsymbol{x}}$. As in the one-dimensional case $D_p = ||\Delta S||_2$, using Eq. 5, it holds $\frac{\partial D_p}{\partial \boldsymbol{x}} = \frac{D_p}{\sqrt{m}\,\text{RMSE}}$.

## A.2 EXPERIMENTAL ILLUSTRATION OF OBSERVATION 1

To illustrate Observation 1 we evaluate the correlation between $\frac{D_p}{\text{RMSE}}$ and the probability of the natural images. The results of the experiment will depend on (a) how closely the considered $D_p$ describes human perception, and (b) how closely human perception actually equalizes $p(\mathbf{x})$. In any case, note that this correlation should be compared to the *zero correlation* obtained if one naively assumes that the perceptual distance is just the RMSE. The illustrations consider (a) two representative measures of perceptual distortion: the Normalized Laplacian Perceptual Distance (the NLPD (Laparra et al., 2016)) and the classical Multi-Scale SSIM (Wang et al., 2003), and (b) a recent model for the probability of natural images, the PixelCNN++ (Salimans et al., 2017)[2]. Here we take images, $\mathbf{x}$, from the CIFAR-10 dataset (Krizhevsky et al., 2009) which consists of small samples that we can analyze according to the selected image model. We corrupt the images with additive Gaussian noise such that $\Delta\mathbf{x} \sim \mathcal{N}(0, \sigma^2)$ for a wide range of noise energies. A pretrained PixelCNN++ model is used to estimate $\log(p(\mathbf{x}))$ of all original images.

Figure 6 shows the correlations, $\rho(\frac{D_p}{\text{RMSE}}, p(\mathbf{x}))$, together with a convenient reference to check the quality of the selected perceptual distances for natural images: we also show the correlation of NLPD and MS-SSIM with the naive Euclidean distances $\rho(D_p, \text{RMSE})$. In these figures, natural images live at the left end of the horizontal axis (uncorrupted samples $\mathbf{x}$). Images with progressively higher noise levels depart from naturalness. The right end of the scale corresponds to highly corrupted and hence non-natural images. For natural images (those with negligible distortion), both perceptual distances are not correlated with RMSE (RMSE is not a plausible perceptual distances). For natural images the correlation considered in the Observation 1 is high (red/gray curves). For $\sigma = 1$, the Spearman correlation is 0.51 for NLPD and 0.63 for MS-SSIM compared to RMSE (which would be $\rho(\frac{\text{RMSE}}{\text{RMSE}}, p(\mathbf{x})) = 0$). The correlation between the perceptual distances and the Euclidean RMSE increases for non-natural images, but note also that the relation between $\frac{D_p}{\text{RMSE}}$ and $p(\mathbf{x})$ decreases for

---

[2]https://github.com/pclucas14/pixel-cnn-pp under MIT License

progressively non-natural images.

These results indicate a strong connection between the sensitivity of perceptual distances (either NLPD or MS-SSIM) and the distribution of natural images, as opposed to the sensitivity of Euclidean RMSE, which is not related to the data.

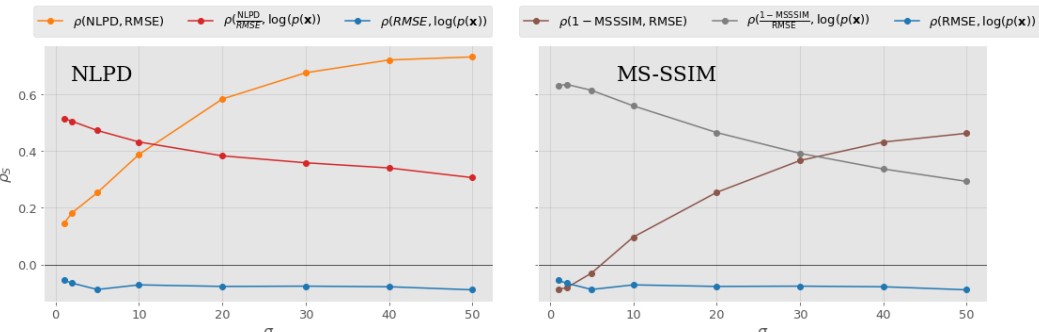

Figure 6: **Illustration of Oberservation 1.** Spearman correlations $\rho_S$ between the sensitivity of the perceptual distances NLPD and MS-SSIM and $\log(p(\mathbf{x}))$ (in red/gray). Distances are computed between $\mathbf{x}$, and a distorted version with additive Gaussian noise, $\mathbf{x}+\Delta\mathbf{x}$, with deviation $\sigma$. Correlation of RMSE with perceptual distortions (in orange/brown) and of RMSE with $\log(p(\mathbf{x}))$ (in blue) are included for comparison. MS-SSIM is a similarity index, so 1-(MS-SSIM) is a distortion measure.

### A.3 DETAILS OF CIFAR 10 EXPERIMENTS

Fig .7 shows how $\frac{\text{NLPD}(\mathbf{x}_1,\mathbf{x}_2)}{\text{RMSE}}$ and $\frac{1-\text{MS-SSIM}(\mathbf{x}_1,\mathbf{x}_2)}{\text{RMSE}}$ vary with $\log(p(\mathbf{x}))$ for each specific image and the two extreme situations of very low and very high noise variance. It can be seen that for low variance, $\sigma = 1$, the probability and the fraction of the perceptual metric are more related than for high variance. This is coherent with the idea of the Observation 1.

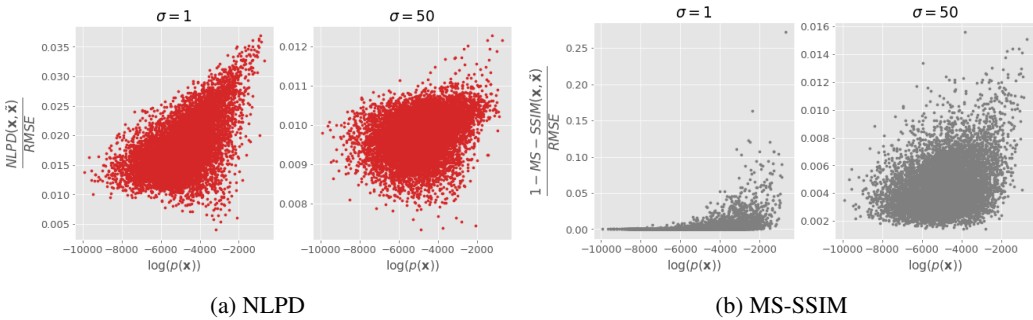

Figure 7: Scatter plots showing $\log(p(\mathbf{x})$ and perceptual distances NLPD and MS-SSIM between images from CIFAR-10 with additive Gaussian noise for $\sigma = 1, 50$

Fig. 8 shows the distribution of $\log(p(\mathbf{x} + \mathcal{N}(0,\sigma^2)))$ for $\sigma = \{0, 1, 2, 5, 10, 20, 30, 40, 50\}$. A pretrained PixelCNN++ model was used to evaluate $\log(p(\mathbf{x}))$. It is clear that when more noise is introduced in the images the probability of being a natural image decrease (As expected).

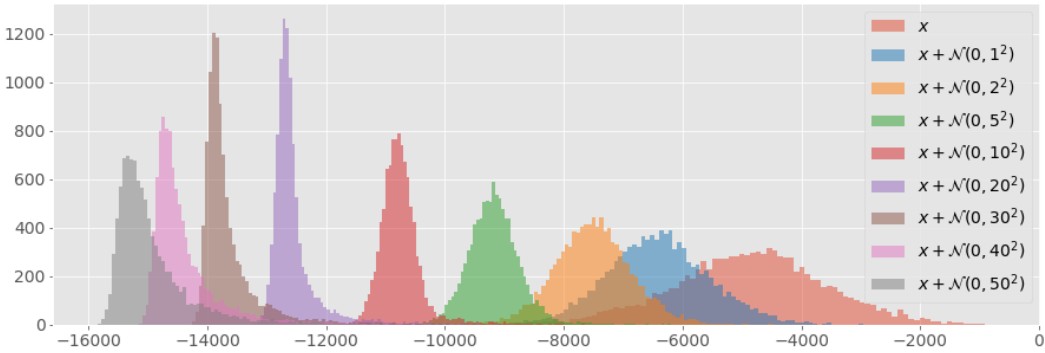

Figure 8: Distributions of $\log(p(\mathbf{x} + \mathcal{N}(0, \sigma^2)))$ for various $\sigma$ where $\log(p(\mathbf{x}))$ is estimated using PixelCNN++ model.

## B  SELF-RECONSTRUCTION DISTANCE AND SCORE MATCHING IN AUTOENCODERS

In (Vincent, 2011) for denoising autoencoders and in (Alain & Bengio, 2014) for contractive autoencoders it is stated that:

$$\mathbf{x} - \hat{\mathbf{x}} = \frac{\partial log(p(\mathbf{x}))}{\partial \mathbf{x}} \tag{7}$$

or equivalently $\mathbf{x} - \hat{\mathbf{x}} = \frac{1}{p(\mathbf{x})} \frac{\partial(p(\mathbf{x}))}{\partial \mathbf{x}}$. From this last formula we can see already that the difference between the original data and the reconstructed data is proportional to the inverse of the probability, as stated in Obs. 2 (just ignoring the term $\frac{\partial(p(\mathbf{x}))}{\partial \mathbf{x}}$ for now).

In particular for the Gaussian distribution we have that the score matching:

$$\frac{\partial log(p(\mathbf{x}))}{\partial \mathbf{x}} = -2\frac{x - \mu}{\sigma^2}, \tag{8}$$

has a monotonical relation with the inverse of the probability (see Fig. 9).

Score matching has been proven to be more general than the Gaussian noise scenario in denoising autoencoders (Alain & Bengio, 2014), and our observation (based on Eq. 2) is even more general than this because Eq. 4 is not attached to any particular noise/metric. Our observation 2 does not have the same analytical expression than the score matching result, but we see it is consistent with the score matching solution in the high probability regions (around the mode of the PDF, see Fig. 9).Our observation observation for $D_s$ and the score matching result only differ in the low probability regions. However there are several things to take into consideration. The obvious one is that we are going to have few data points that are going to fall in low probability region. This is important not only in evaluation terms but also during the training procedure of the autoencoder. Note that the autoencoder is not going to be properly trained in these regions and therefore it will tend to interpolate what it has learn in medium probability regions. For instance the autoencoder is going to try to project the data that from very low probability regions into the manifold, therefore the distance between the original data and the reconstructed data ($||\mathbf{x} - \hat{\mathbf{x}}||_2$) is going to be high (although the theory, Eq. 7 infers that it should be extremely low).

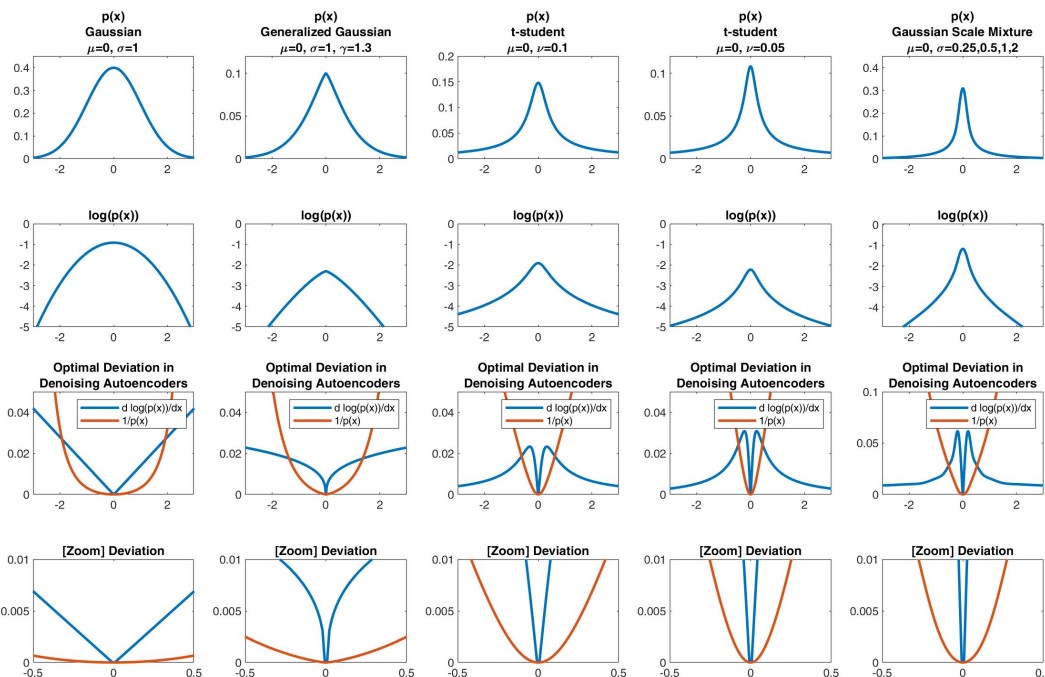

Figure 9: Magnitude of the deviation introduced by denoising autoencoders in a Gaussian PDF (first column) and in different PDFs proposed to model natural images (rest of the columns). The score matching, $\frac{\partial log(p(x))}{\partial x}$, is correlated to $\frac{1}{p(x)}$ in the mode of the distributions. Zoomed-in in the last row.

## C  2D EXPERIMENTS

Throughout the paper, a 2D example is used to explore the various observations in a simplified setting. Here, details can be found on the exact experimental setup.

We define the probability density function of our 2D Student-t distribution as

$$\boldsymbol{\mu} + \boldsymbol{\sigma}\mathbf{t}_\nu$$

where we use $\boldsymbol{\mu} = \left(\begin{smallmatrix} 0 \\ 0 \end{smallmatrix}\right)$, $\boldsymbol{\sigma} = \begin{bmatrix} 0.5 & 0 \\ 0 & 0.2 \end{bmatrix}$ and $\mathbf{t}_\nu$ are samples from a 2D Student-t distribution with degrees of freedom $\nu = 2$. A dimension with high variance and one with low variance were chosen to mimic the low and high frequency bands in natural images.

A 3-layer multilayer perceptron (MLP) is used for $e$ and $d$, where the dimensions through the network are $[2 \rightarrow 100 \rightarrow 100 \rightarrow 2]$ for $e$ and the reverse for $d$. The quantisation used is rounding to the nearest integer, which is approximated during training with additive uniform noise as in uniform scalar quantiastion the gradients are zero almost everywhere. Fig. 10 shows the architecture of the autoencoder. Softplus activations are used on all hidden layers but omitted from the final layers to not restrict the sign of the representation or reconstruction. The network is optimized using Eq. 3, using different $\lambda$ values to achieve different rates. Adam optimizer was used, with a learning rate of 0.001, a batch size of 4096 and 500,000 steps where for each step a batch is sampled from the distribution. The large batch size was to account for the heavy tailed distribution used. Fig. 11 shows a) samples from the 2D Student-t distribution b) density of the distribution and c) resulting compression to a rate of 2.65 bits per pixel (bpp) where the dots represent code vectors and lines represent quantisation boundaries.

In Sec. 5.1, we hypothesize that one can cause a model to focus on high probability regions by including the probability in the loss function. In order to see the effect of including probability in the distortion term, similar to what we hypothesize perceptual metrics are doing for images, we use $p(\mathbf{x})^\gamma \cdot \mathcal{D}$ as the distortion term, with $\gamma = \{-0.1, 0, 0.1\}$. $\gamma$ was chosen to have a small magnitude to guarantee stable training and to avoid the model collapsing to one code vector assigned at the center

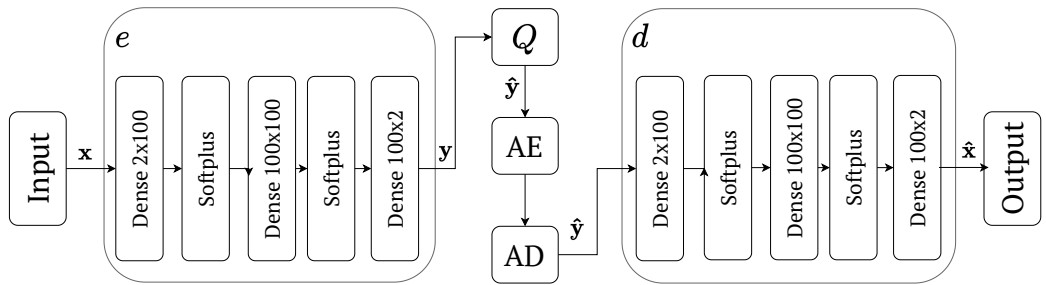

Figure 10: The network architecture for the 2D example where $e$ is the encoder, $d$ the decoder, $Q$ is the quantisation step, AE is a arithmetic encoder and AD is the arithmetic decoder. The quantisation $Q$ used is rounding to the nearest integer, which is approximated by additive uniform noise at training time.

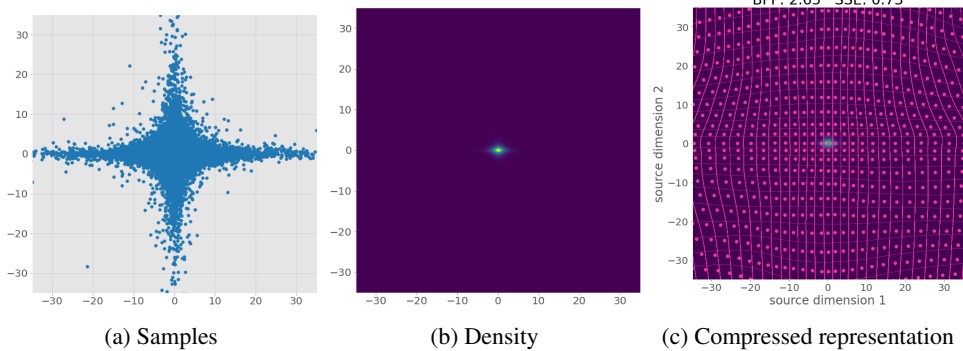

(a) Samples       (b) Density    (c) Compressed representation

Figure 11: The 2D Student-t distribution used throughout the paper and one compressed representation found by minimizing Eq. 3. For c) the lines represent quantisation boundaries (within a bin, all points are compressed to the same point by $e$ and the dots represent code vectors (where these points are projected to by $d$.

of the distribution. Thus the loss functions to optimize for are

$$\mathcal{L}_1 = \mathcal{R} + \lambda \mathcal{D} \tag{9}$$

$$\mathcal{L}_2 = \mathcal{R} + p(\mathbf{x})^{0.1} \cdot \lambda \mathcal{D} \tag{10}$$

$$\mathcal{L}_3 = \mathcal{R} + p(\mathbf{x})^{-0.1} \cdot \lambda \mathcal{D} \tag{11}$$

where the $\lambda$ parameters are tuned so that the networks have similar rate ranges and for $\mathcal{D}$ the sum of squared errors (SSE) is used. Fig. 12 shows examples of compression resulting from optimizing each of the 3 loss functions. The figure clear shows that when multiplying the distortion by $p(\mathbf{x})^{0.1}$ the quantisation bins and code vectors concentrate on the support of the distribution, whereas multiplying by $p(\mathbf{x})^{-0.1}$ enforces a more uniform distribution of code vectors.

In Sec. 5.1 we observe the performance gain in using $p(\mathbf{x})$ in the loss function. For a network, given a set of points to evaluate on we take the general performance to be the distortion over the rate,

$$\mathcal{P} = \frac{\mathcal{D}}{\mathcal{R}} \tag{12}$$

Note that this is a sort of normalization in order to compare networks that compress to different rates. For the 2-D example, the gain in performance in including $p(\mathbf{x})^{0.1}$ in the distortion term can then be defined as

$$\frac{\mathcal{P}_1}{\mathcal{P}_2} \tag{13}$$

where $\mathcal{P}_1$ is the performance of a network trained with loss $\mathcal{L}_1$ (Eq. 9) and $\mathcal{P}_2$ is the performance of network trained with loss $\mathcal{L}_2$ (Eq. 10). For the experiment in Sec. 5.1, the relative performance is calculated for points samples along the positive x-axis, $(x, y) \in ([0, 35], 0)$ and the results shown in Fig. 4.

In Sec. 5.2, we show that autoencoders can be trained without data. Instead, provided with direct

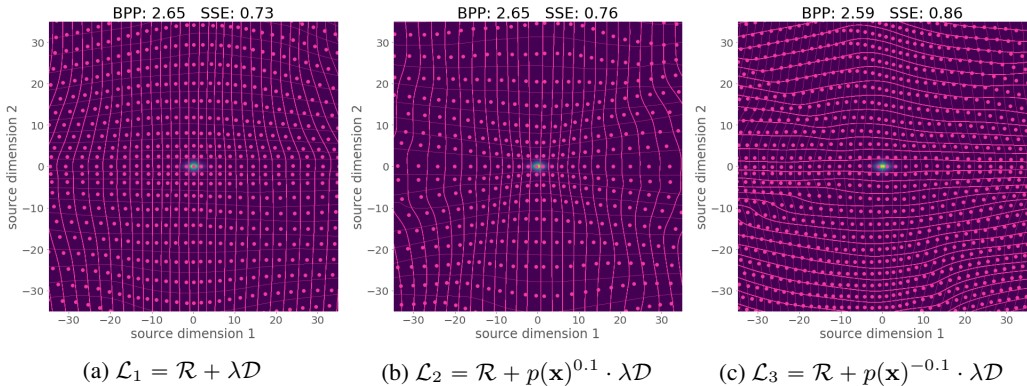

(a) $\mathcal{L}_1 = \mathcal{R} + \lambda\mathcal{D}$     (b) $\mathcal{L}_2 = \mathcal{R} + p(\mathbf{x})^{0.1} \cdot \lambda\mathcal{D}$     (c) $\mathcal{L}_3 = \mathcal{R} + p(\mathbf{x})^{-0.1} \cdot \lambda\mathcal{D}$

Figure 12: Resulting compression when using the 2D Student-t distribution and including the probability distribution in the loss function. b) is seen as including information about the distribution in the loss function and c) is seen as removing it. BPP is bits per pixel (the rate) and SSE is sum square errors (distortion) evaluated on a validation set.

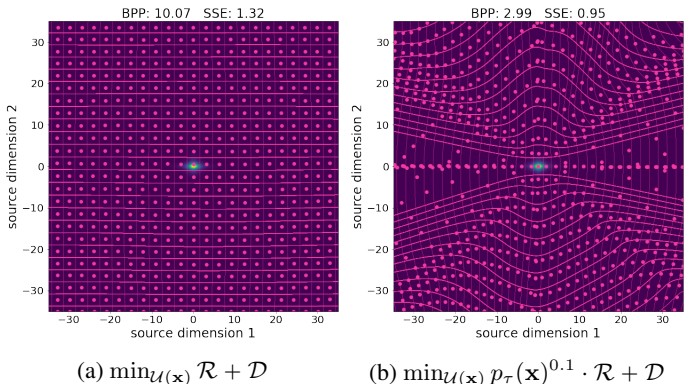

(a) $\min_{\mathcal{U}(\mathbf{x})} \mathcal{R} + \mathcal{D}$       (b) $\min_{\mathcal{U}(\mathbf{x})} p_\tau(\mathbf{x})^{0.1} \cdot \mathcal{R} + \mathcal{D}$

Figure 13: Resulting compression when using a 2D uniform distribution across the space, where left) optimized the rate-distortion equation over the uniform distribution and right) optimized for the probability belonging to 2D Student-t weighting the rate-distortion equation over the uniform distribution.

access or a proxy to the probability distribution, one can achieve reasonable results when training over samples from a uniform distribution. Fig 13b shows the resulting compression using the 2D Student-t distribution where the loss that has been minimized is $\min_{\mathbf{x} \sim \mathcal{U}} \mathcal{L} = p_\tau(\mathbf{x})^{0.1} \cdot (\mathcal{R} + \lambda\mathcal{D})$, where $p_\tau(\mathbf{x})$ denotes the probability of point $\mathbf{x}$ belonging to the 2D Student-t distribution. This training procedure requires no data sampled from the Student-t distribution, only access to evaluations of the probability and results in code vectors assigned to the support of the distribution, namely the x-axis which has the highest variance for the 2D student distribution. The loss functions contains information on the probability distribution and thus does not require samples.

In Fig.4 we show the curves of the relative performance of the 2D example. In order to facilitate the visualization we use a 20 degree polynomial fitting to soft the curve. In Fig.14 we show the result after and before the fitting. Note that the fitting makes sense since there will be a variability between the reconstruction error of close points since point close to the code vectors will be reconstructed much better than points from the same region but far from the code vector.

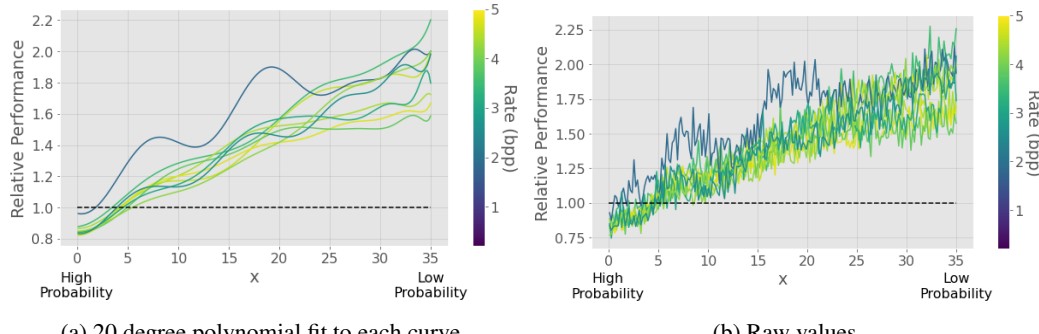

(a) 20 degree polynomial fit to each curve.      (b) Raw values.

Figure 14: Relative performance of networks for samples along a line through the support of the respective distributions. Left: networks trained with $\mathcal{D} = p(\mathbf{x}) \cdot ||\mathbf{x}_1 - \mathbf{x}_2||_2^2$ in Eq. 3 divided by performance of networks trained with $\mathcal{D} = ||\mathbf{x}_1 - \mathbf{x}_2||_2^2$ on the 2D Student-t and evaluated using samples along $x$-axis. Left) shows 20 degree polynomial fit to each network and right) shows the raw values.

## D IMAGES

### D.1 COMPRESSION AUTOENCODER

In Sec. 4 & 5.1 pretrained autoencoders are used, namely the factorized prior model from (Ballé et al., 2018). This model has layers of convolution operations and generalized divisive normalization activations, the architecture is show in Fig 15.

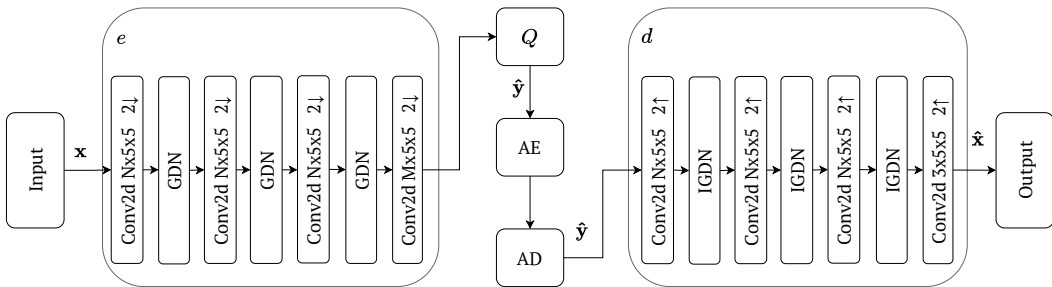

Figure 15: Architecture for networks used in Sec. 4 & 5.1 which is the the factorized prior model from (Ballé et al., 2018) where $e$ is the encoder, $d$ the decoder, $Q$ is the quantisation step, AE is a arithmetic encoder and AD is the arithmetic decoder. The quantisation $Q$ used is rounding to the nearest integer, which is approximated by additive uniform noise at training time. GDN denotes a generalized divisive normalization activation, and Conv2d is a 2-d convolution operation. The convolution parameters are filters × kernel height × kernel width - down- or upsampling stride. For the 5 lower bit rates, $N = 128$, $M = 192$ and for higher rates $N = 192$, $M = 320$.

We take the performance gain again as Eq. 13, where $\mathcal{P}_1$ is the performance of a network trained with using MSE as the distortion $\mathcal{D}$ and $\mathcal{P}_2$ is the performance of a network trained with using MS-SSIM as distortion (akin to $p(\mathbf{x}) \cdot \mathcal{D}$ in the 2D case).

### D.2 SAMPLING THROUGH DISTRIBUTION OF IMAGES

Starting from the center of the distribution (high probability), a direction on the support of the distribution is chosen and samples are generated along this line. For the 2D distribution, we use $(x, y) \in ([0, 35], 0)$ to generate samples along the x-axis. Fig. 4 shows the proportion defined earlier for samples generated along the x-axis. With the 2D example we have explicit access to the probability distribution, a luxury we are not afforded when it comes to images due to the probability distribution being intractable. Drawing a line through the distribution of natural images in the same way as in the 2D example would be ideal, although infeasible. However, it has been shown that

lower-contrast images are more likely (Frazor & Geisler, 2006) and as such we use contrast as an axis to sample through the distribution of natural images. For each image in the Kodak dataset a low-contrast and high-contrast version is generated. Samples are then taken between a linear interpolation between the low contrast and original, and the high contrast and original for all images in the Kodak dataset (Kodak, 1993). For each image in the Kodak dataset, a high contrast version $\mathbf{x}_{high}$ and a low contrast version $\mathbf{x}_{low}$ are created. Linear interpolation is used to get a gradual shift from low contrast-original-high contrast.

$$\hat{\mathbf{x}} = \begin{cases} (1 - \alpha) \cdot \mathbf{x} + \alpha \cdot \mathbf{x}_{\text{high}} & \text{if } \alpha > 1 \\ (1 + \alpha) \cdot \mathbf{x} - \alpha \cdot \mathbf{x}_{\text{low}} & \text{if } \alpha \leqslant 1 \end{cases} \tag{14}$$

for $\mathbf{x}$ images in the Kodak dataset and $\alpha \in [0, 2]$, where $\alpha = 0$ denotes the original image. 200 $\alpha$

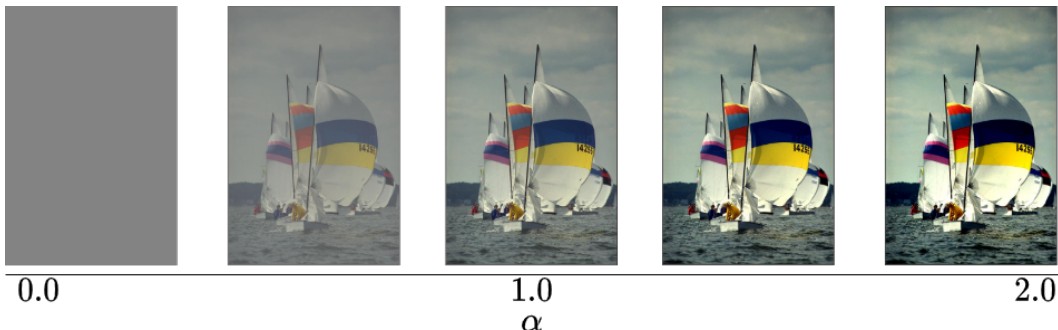

0.0          1.0          2.0

$\alpha$

Figure 16: Example of altering the contrast of an image from the Kodak Image dataset varying $\alpha$ in Eq. 14

values are sampled. Fig. 16 shows an example for samples generated for one image. We compare networks with probability included in the loss function, in the form of perceptual distances, to those without. The ratio defined earlier will be used, where $\mathcal{P}_1$ is the distortion/rate for networks optimized with MS-SSIM and $\mathcal{P}_0$ is distortion/rate for networks optimized for MSE. The ratio defined in Eq. 13, where the numerator denotes networks optimized using perceptual metric MS-SSIM and the denominator denotes networks optimized for MSE. All networks were pretrained and taken from the Tensorflow Compression package. Fig. 4 shows the ratio of performance as we vary $\alpha$.

## E  ENTROPY LIMITED AUTOENCODERS

For ease of training, in Sec. 5.2 we simplify the rate-distortion loss function by setting an upper bound on the entropy and just minimizing for the distortion as in (Ding et al., 2021; Agustsson et al., 2019).

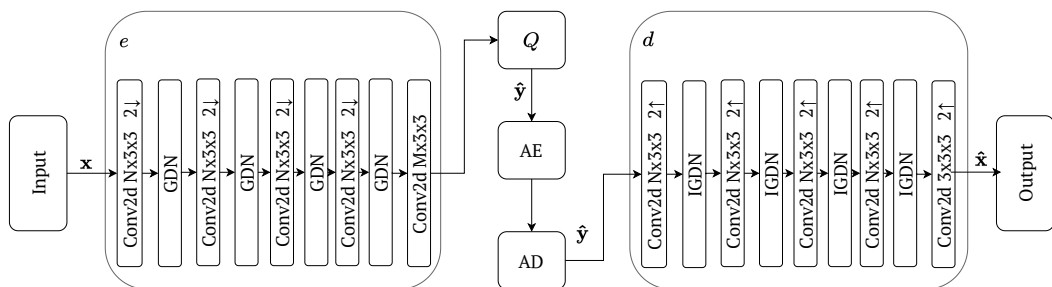

Figure 17: Architecture of networks used in Sec. 5.2 where $e$ is the encoder, $d$ the decoder, $Q$ is the quantisation step, AE is a arithmetic encoder and AD is the arithmetic decoder. The quantisation $Q$ used is rounding to the nearest center defined by $L$, which is approximated by Eq. 15 at training time. GDN denotes a generalized divisive normalization activation, and Conv2d is a 2-d convolution operation. The convolution parameters are filters × kernel height × kernel width - down- or upsampling stride. For all rates $N = 128$, $M = 64$

Setting an upper bound on the entropy of the encoding with $L$ centers $c_1, ..., c_L$, the soft differentiable

approximation is

$$\hat{y}_i = \sum_{j=1}^{L} \frac{\exp(-s(y_i - c_j)^2)}{\sum_{k=1}^{L} \exp(-s(y_i - c_k)^2)} c_j \tag{15}$$

where $s$ is the quantisation scale parameter which we fix to 1. Given that we know the dimensionality of $y$ and a maximum of $L$ integers that can be represented, an upper bound on the entropy can be obtained;

$$H(y_i) \leqslant \frac{W \times H}{2^n \cdot 2^n} \cdot m \cdot \log_2(L) \tag{16}$$

where $[H, W]$ are dimensions of the image, $n$ is the number of downsampling layers you have in the network with stride 2, $m$ is the number of channels in your embedding and $L$ is the number of centers the embeddings are rounded to.

For all experiments in Sec. 5.2, an architecture is used with 5 convolutional layers and 4 GDN layers in both $e$ and $d$. Fig. 17 shows the entire architecture. In this case, the quantisation is performed by rounding values to the centers defined by $L$. Table 1 shows the results of training these autoencoders using different training distributions, a distribution of natural images $P(\mathbf{x})$ and a uniform distribution $\mathcal{U}(\mathbf{x})$ and using different loss functions, perceptual metrics NLPD and MS-SSIM and the MSE. 2 networks for each distribution and loss was trained, one compressing to an upper bound of 0.25bpp and another 0.5bpp. Probably the most interesting part of this table is shown in the PSNR column. Note how for high entropy the network trained to minimize NLPD gets better performance in PSNR than the one trained for MSE (which is basically PSNR). This effect matches with the *double counting effect* proposed in sec. 5. The idea is that minimizing the NLPD is related with minimizing the expected MSE which is what is at the end being evaluated. This effect is much more visible in the same column when using the uniform distribution for training. In this case the improvement when using NLPD is clear regard the MSE. This can only be the case if the NLPD has some properties of the distribution of the natural images.

Table 1: Evaluating autoencoders trained with both uniform distribution $\mathcal{U}(\mathbf{x})$ and the OpenImages dataset (Krasin et al., 2017) $\mathcal{P}(\mathbf{x})$ at two different rates. Reported are the PSNR, MS-SSIM and NLPD evaluated for all networks. These networks use the approximate in Eq. 15. Bold values indicate the best value for each evaluation metric at a certain rate (bpp) with a certain distribution.

| Training Distribution | Bits per pixel (bpp) | Distortion Loss | PSNR | MS-SSIM | NLPD |
|---|---|---|---|---|---|
| $P(\mathbf{x})$ | 0.25 | MSE | **29.39** | 0.9755 | 2.615 |
| | | MS_SSIM | 26.91 | **0.9754** | 4.080 |
| | | NLPD | 27.85 | 0.9655 | **2.510** |
| | 0.5 | MSE | 29.73 | 0.9777 | 2.450 |
| | | MS_SSIM | 27.53 | **0.9827** | 4.139 |
| | | NLPD | **30.13** | 0.9809 | **1.792** |
| $\mathcal{U}(\mathbf{x})$ | 0.25 | MSE | 13.17 | 0.4555 | 16.50 |
| | | MS-SSIM | 18.92 | **0.7483** | 10.43 |
| | | NLPD | **21.21** | 0.7396 | **8.090** |
| | 0.5 | MSE | 13.15 | 0.4396 | 16.89 |
| | | MS-SSIM | 18.90 | **0.7659** | 10.37 |
| | | NLPD | **20.59** | 0.7335 | **9.132** |

Fig. 18 shows visual results of the effect in the reconstruction in an image from the Kodak dataset using each of the networks compressing to 0.25bpp and 0.5bpp and trained using natural images as input samples. i.e. $p(\mathbf{x})$. In this case results when using a non perceptual metric as the MSE or a perceptual metric (NLPD or MS-SSIM) are very similar. Even than the correlation of NLPD and MS-SSIM with human perception is much higher than the one of MSE. This is because optimizing to minimize the expected MSE over natural images impose properties of the distribution of the natural images in the autoencoder (as shown in sec.4) and this translates on a kind of perceptual behavior (as shown in sec.2).

On the other hand Fig. 19 shows the reconstruction results when training the autoencoders using data coming from a uniform distribution, i.e. the networks do not see any natural image during the training. However the networks trained using NLPD and MS-SSIM obtain a good performance when

evaluated in a natural image. While, as expected, the network trained to minimize MSE obtains a very bad result.

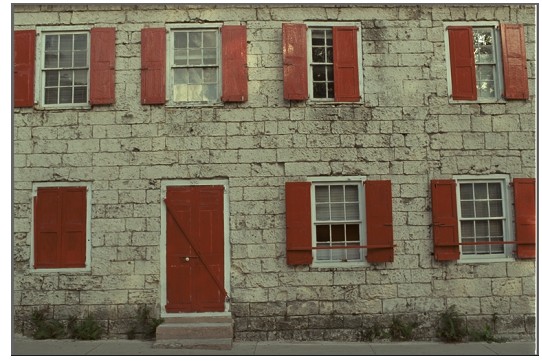

(a) Original

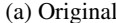

(b) $\min_{p(\mathbf{x})} ||\mathbf{x} - \hat{\mathbf{x}}||_2$ at 0.25 bpp

(c) $\min_{p(\mathbf{x})} ||\mathbf{x} - \hat{\mathbf{x}}||_2$ at 0.5 bpp

(d) $\min_{p(\mathbf{x})} \text{NLPD}(\mathbf{x}, \hat{\mathbf{x}})$ at 0.25 bpp

(e) $\min_{p(\mathbf{x})} \text{NLPD}(\mathbf{x}, \hat{\mathbf{x}})$ at 0.5 bpp

(f) $\min_{p(\mathbf{x})} 1 - \text{MS-SSIM}(\mathbf{x}, \hat{\mathbf{x}})$ at 0.25 bpp

(g) $\min_{p(\mathbf{x})} 1 - \text{MS-SSIM}(\mathbf{x}, \hat{\mathbf{x}})$ at 0.5bpp

Figure 18: The reconstruction of image 1 from Kodak dataset from the various networks trained to compress to a maximum entropy bits per pixel (bpp) specified optimized using the OpenImages dataset (Krasin et al., 2017).

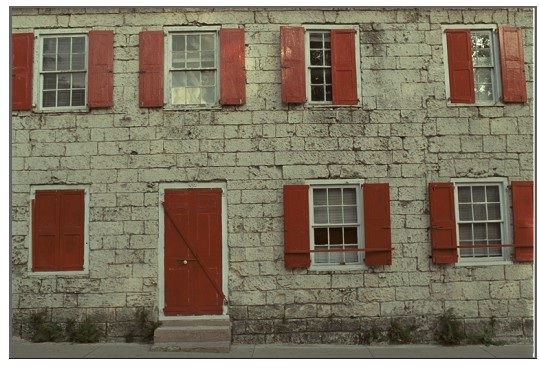

(a) Original

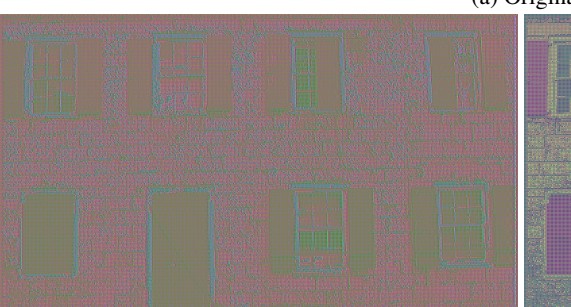
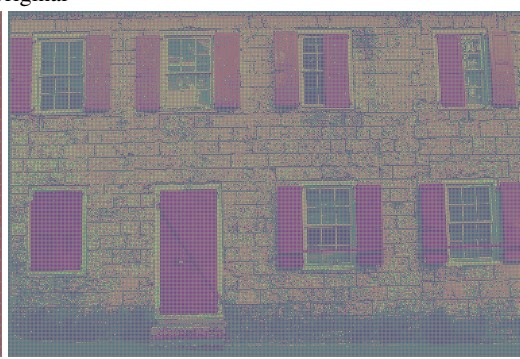

(b) $\min_{\mathcal{U}(\mathbf{x})} ||\mathbf{x} - \hat{\mathbf{x}}||_2$ at 0.25 bpp       (c) $\min_{\mathcal{U}(\mathbf{x})} ||\mathbf{x} - \hat{\mathbf{x}}||_2$ at 0.5 bpp

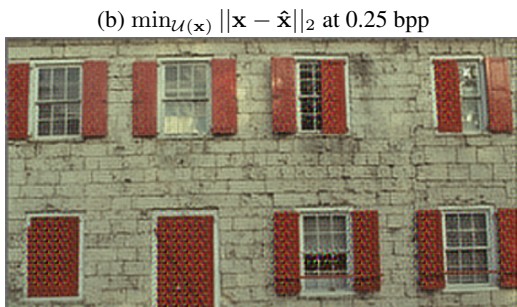
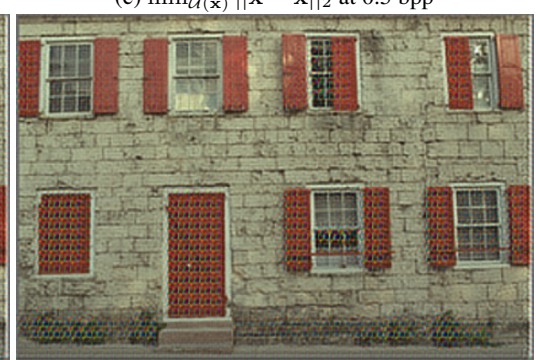

(d) $\min_{\mathcal{U}(\mathbf{x})} \text{NLPD}(\mathbf{x}, \hat{\mathbf{x}})$ at 0.25 bpp       (e) $\min_{\mathcal{U}(\mathbf{x})} \text{NLPD}(\mathbf{x}, \hat{\mathbf{x}})$ at 0.5 bpp

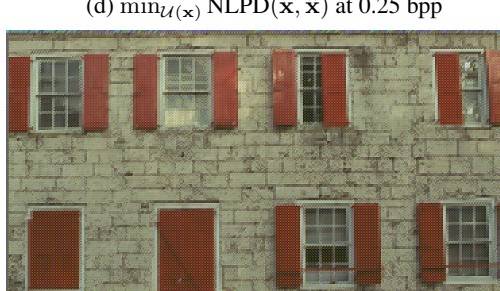
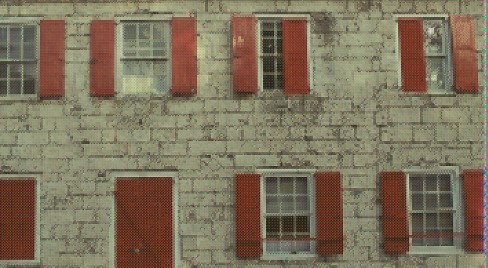

(f) $\min_{\mathcal{U}(\mathbf{x})} 1 - \text{MS-SSIM}(\mathbf{x}, \hat{\mathbf{x}})$ at 0.25 bpp       (g) $\min_{\mathcal{U}(\mathbf{x})} 1 - \text{MS-SSIM}(\mathbf{x}, \hat{\mathbf{x}})$ at 0.5bpp

Figure 19: The reconstruction of image 1 from Kodak dataset from the various networks trained to compress to a maximum entropy bits per pixel (bpp) specified optimized using random uniform noise.

### E.1 TRAINING WITH SMALL BATCH SIZES

In Sec. 5.2 we explore what happens when one does not have direct access to the data. A more relaxed version of this is having access to a small amount of data at a time; using small batches of images.

For small batch sizes, the minibatch estimate of the gradient over the whole set of images will have higher variance and more likely to be effected by outliers. From Sec. 5.1 we can see that using a perceptual distance is similar to multiplying by the probability, which would weight gradients calculated from outliers less. In theory, this should lead to a more accurate estimate over several minibatches for a perceptual distance rather than MSE. For stochastic gradient descent, minibatches of data are used to estimate the expected loss and aim to minimize the generalization error (Goodfellow et al., 2016). The exact gradient of the generalization error with loss function $\mathcal{L}$ over model $f$ is given by

$$\mathbf{g} = \sum_{\mathbf{x}} p(\mathbf{x}) \nabla_f \mathcal{L}(f(\mathbf{x}), \mathbf{x}). \tag{17}$$

In gradient descent, $\mathbf{g}$ is estimated by sampling a batch of size $m$ from data distribution $p(\mathbf{x})$ and computing the gradient of the loss with respect to $f$

$$\hat{\mathbf{g}} = \frac{1}{m} \nabla_f \sum_{i}^{m} \mathcal{L}(f(\mathbf{x}_i), \mathbf{x}_i). \tag{18}$$

Gradient updates are then performed using this $\hat{\mathbf{g}}$. Given this estimator, we can quantify the degree of expected variation in the estimated gradients using the standard error of mean

$$\text{SE}(\hat{\mathbf{g}}_m) = \sqrt{Var\left[\frac{1}{m} \nabla_f \sum_{i}^{m} \mathcal{L}(f(\mathbf{x}_i), \mathbf{x}_i)\right]} = \frac{\sigma}{\sqrt{m}} \tag{19}$$

where $\sigma^2$ is the true variance of the gradient of the loss function for point $\mathbf{x}_i$. Given that the true variance $\sigma^2$ is unknown, if batch size $m$ is small then a loss function with low variance will give us a better estimate of the generalization error and therefore a better estimate of the gradient too. When performing stochastic gradient descent (SGD) using extremely small batch size, e.g. 1, the gradients have higher variance as your estimation of your weight updates become more inaccurate when attempting to estimate the ideal weight update for the entire training set. For example, with a batch size of 1 weight updates that come from outliers have the potential to move the weight vector far away from the optimum value it was in the process of meeting. In order to reduce the effect of outliers, one would need explicit access to the underlying data distribution. This is exactly the type of regularization that perceptual distances can perform. Given that they are proportional to the probability distribution of natural images (Section 2), the perceptual distance between an outlier and it's reconstruction will be weighted less than images coming from high density regions, and thus achieve a lower standard error for the estimator of the generalization error (Eq. 19) compared to Euclidean distances like MSE.

Tho illustrate this effect we performed an experiment where we train the same network using a batch size of 1 one to minimize MSE and one is trained to minimize NLPD. We evaluate them on the expected MSE over the Kodak image dataset (Kodak, 1993). This means that for the network optimized for NLPD, the loss function we are minimizing and the function we use to measure generalization error are different. This quantity can be evaluated after every batch of size 1 and Fig. 20 reports the ratio of generalization error over the test set for the network trained for NLPD over MSE. Random seeds are fixed so that the two networks are given the exact same initialization and same order of images. This autoencoder acts purely as dimensionality reduction and has no quantisation in the embedded domain. It follows the same architecture as in Fig. 17 without the quantisation, AE and AD steps. 5 different random seeds were used, i.e. different network initialization and different order of training images, and reported is the mean and standard deviation across runs. The true expected MSE over the test dataset is given as $\mathbb{E}_{test} MSE(\mathbf{x}, \hat{\mathbf{x}})$. This quantity is evaluated after every batch of size 1 and Fig. 20 reports the ratio of the expected MSE over the test set for the network trained for NLPD over MSE.

On average, the network trained using NLPD achieves a lower expected MSE than the network optimized for MSE, even though we are evaluating using the same function as it's loss. This ability to estimate the test set expected MSE is indicative of NLPD being getting a better estimate of the gradient of the generalization error.

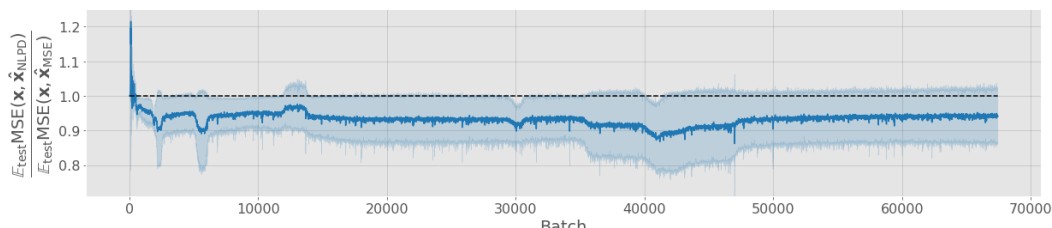

Figure 20: Gain in using NLPD over MSE as a loss function evaluated in terms of MSE loss on test set (Kodak dataset) using batch size of 1 and a small learning rate, fixing random seeds. $\hat{\mathbf{x}}_{\mathrm{NLPD}}$ denotes the reconstruction of $\mathbf{x}$ with a network optimized for NLPD, and $\hat{\mathbf{x}}_{\mathrm{MSE}}$ for a network optimized for MSE. The mean (solid line) and standard deviation (solid fill) was taken over 5 runs with different random seeds, i.e. different network initialization and training image ordering. The dashed line represents if the two networks had the same expected MSE on the test set.

