# OpenReview forum: "On the relation between statistical learning and perceptual distances"
_ICLR.cc/2022/Conference — ICLR 2022 Spotlight_

### Official Review · Reviewer_Rmvg · 2021-11-01

**Correctness:** 4
**Technical Novelty And Significance:** 4
**Empirical Novelty And Significance:** 3
**Recommendation:** 8
**Confidence:** 2

**Main Review:**

While the paper is dense, I found it nevertheless to be well structured and the figures to be generally helpful. While at first I thought that the result might be trivial (the human visual system compresses natural input, so it makes sense that autoencoders and perceptual distances are related), I believe that the paper draws out the value of making these connections explicit --- though I am not an expert in the theoretical side of this field. I have minor comments.

- The observations presented are limited to small image differences $\delta$. I would be interested to know (1) if the theory can be extended to estimate bounds on $\delta$ in some meaningful dimension and (2) the authors' speculations on whether the assumption of perceptual distances as metric spaces breaks down when $\delta$ is large (see e.g. Tversky, 1977, *Features of Similarity*), as in the suprathreshold regime comparing different types of image manipulations.

- Could the "double-counting" effect be mitigated by reweighting the relative contributions of perceptual loss and training?

- Figure 3: why is the RMSE correlation with MOS (dashed line) flat as a function of bpp?

- Figure 4: there seems to be quite some kernel smoothing happening here, making it unclear how much data this actually represents.

- P. 8 typo "for perceptual distances we are ,"

- A relevant reference for the introduction linking efficient coding and compressed unsupervised representations is: Storrs, K. R., Anderson, B. L., & Fleming, R. W. (2021). Unsupervised learning predicts human perception and misperception of gloss. *Nature Human Behaviour.*

**Summary Of The Paper:**

This paper presents a mainly theoretical explication of the relationship between natural image statistics and perceptual distances for small image distortions. The paper presents a number of observations linking distances in natural images, autoencoders, and perceptual similarity for humans. The paper finally explores some implications of these observations, including impressive-seeming results on training with no data using perceptual distances as regularizers.


**Summary Of The Review:**

A theoretical explanation of the link between natural image statistics, autoencoders and perceptual distances, with some seemingly important implications. This is not exactly my field so I may be missing relevant background.

---

> ### Author Response · Authors · 2021-11-16
> **Response to Review Rmvg**
>
> Thank you for your review. The typos have already been fixed in the new version.
>
> > The observations presented are limited to small image differences $\delta$. I would be interested to know (1) if the theory can be extended to estimate bounds on $\delta$ in some meaningful dimension and (2) the authors' speculations on whether the assumption of perceptual distances as metric spaces breaks down when $\delta$ is large (see e.g. Tversky, 1977, Features of Similarity), as in the suprathreshold regime comparing different types of image manipulations.
>
> The observations are limited to small $\delta$, which is mainly to achieve that the distorted images still lie on the distribution of natural images. We are not sure how this could be extended to estimate bounds - as it would require explicit knowledge of image probability. With a large delta, we still observe some correlation but it is indeed lower (Fig. 2) as the images become less natural (and less probable).
>
> > Could the "double-counting" effect be mitigated by reweighting the relative contributions of perceptual loss and training?
>
> The double counting effect could be effectively mitigated with explicit access to the image probability. This is similar to what we do in the 2D example when multiplying by $p(\mathbf{x})^{-0.1}$. One could also define a metric which is a weighted mean of the Euclidean and perceptual loss, meaning we can control the weighting depending on the image, however this would require a lot of further work.
>
> > Figure 3: why is the RMSE correlation with MOS (dashed line) flat as a function of bpp?
>
> The RMSE correlation with MOS is not dependent on bpp, but it is a baseline of the RMSE between the original and distorted correlated with MOS. We just wanted to show that the correlations are higher than this baseline.
>
> > Figure 4: there seems to be quite some kernel smoothing happening here, making it unclear how much data this actually represents.
>
>  In Fig. 4 there is kernel smoothing due to the quantisation bins in 2D, however we provide an unsmoothed version in the appendix (Fig. 14).

---

### Official Review · Reviewer_CZqT · 2021-11-02

**Correctness:** 4
**Technical Novelty And Significance:** 1
**Empirical Novelty And Significance:** 2
**Recommendation:** 6
**Confidence:** 5

**Main Review:**

#### Strengths :

* Interesting topic at the cross-road of perception and machine/deep learning (ML/DL)
* Good review of the literature
* Well written and understandable

#### Weaknesses :

* No major contributions
* List of observations that are already known in the ML/DL community
* More like a review paper

### Detailed comments

Overall reading this paper is interesting as it gathers two/three fields: ML/DL, neurosciences and psychophysics. However, I don't really see any contribution to any of these fields.

For ML/DL: most of the observations are already known. No one in the ML/DL community would be surprised to hear that distances between images are correlated with their distribution. Gans, normalizing flows or NeuralODEs are approaches that takle the same goal: mimicking the dataset distribution using a latent space representation.

Neurosciences: No data, nor models are provided.

Psychophysics: No new data is presented. Nothing new is presented about the relation between perceptual distance and human perception.

I fail to understand why the authors introduces so many distances D_s, D_r, D_e and D_in and why it is interesting. In addition, why is it insteresting to state all those observations (that are variant from each others and somehow all result from the efficient coding hypothesis) ?

To finish, on a better note. I think, the authors are working on an interesting topic and I encourage them to pursue. However, I also think this paper is too preliminary. To me the most interesting contribution (which is only a small part of the paper) is the training of auto-encoders with noise and a perceptual similarity distance. Really, is it useful for training with less data ?


**Summary Of The Paper:**

The authors present few relation between statistical learning and perceptual distances. Specifically, they explain that

1. perceptual distances correlate with image likelihoods;
2. auto-encoder latent space induced distances of natural images are correlated with training data probability and human perception;
3. perceptual distances are redundant with Euclidean distances in latent space.


**Summary Of The Review:**

I think it is more a review paper and there are not important enough contributions.

---

> ### Author Response · Authors · 2021-11-16
> **Response to Reviewer CZqT**
>
> Thank you for your comments.
>
> The reviewer states that the paper contains a list of observations that are already known in the community and it is more like a review paper.
> Some of these points may have been done separately in disparate contexts, but we disagree that they have been presented with the aim of linking the three fields from a single point of view as proposed here. However, we are happy to be proven wrong, so we would like the reviewer to provide us with references of these observations linking these three fields already being known.
>
> In fact, ICLR is the perfect venue to start a discussion around the different representations discussed in the paper and hopefully lead to a more formal understanding of the relationship between the three fields.
>
> Specifically, for ML/DL, it is obvious that the models learn some characteristics from the underlying distribution they are trained with and this has been shown in the work done on score matching. However, we have not seen work that explicitly links induced distances to the probability distribution they are trained with and would be very keen to see papers that do so. In particular, with autoencoders, we find it surprising and non-trivial that the self-reconstruction distance is also dependent on the distribution. For psychophysics, previous work has shown the relation between perceptual distances and human perception (actually that is the approach of the older methods to compute perceptual distances), however we would be keen to see papers that have previously quantified the correlation between perceptual distance and image probability, and image probability and human perception.
>
> Evaluating each of the induced distances covers the work of previous publications, which have considered each of these distances individually. To cover our bases, we showed the behaviour over all the distances, as they exhibit different behaviour to one another.
>
> The training of autoencoders with noise is a result of the observations throughout the paper - the interplay between perception, machine learning and image probability. Regarding the usefulness of training with less data - it depends on the practitioner. We show that training on an image dataset with a perceptual loss leads to an overstress in high probability regions. If the practitioner is fine with a loss in performance on less probable images, then they should continue to use a perceptual loss. Training with less data becomes useful when one considers the performance of less probable (but still natural) images.

---

> > ### Comment · Reviewer_CZqT · 2021-11-21
> > **Response**
> >
> >
> > The authors are right that their claims have not been formalized in the literature. However, I maintain that the idea is somehow obvious and I fail to understand how their more precise statements can be useful.
> > The reason why I think it is obvious is that there are many papers using interpolation in latent space :
> >
> > * Bojanowski, P., Joulin, A., Lopez-Pas, D., & Szlam, A. (2018, July). Optimizing the Latent Space of Generative Networks. In International Conference on Machine Learning (pp. 600-609). PMLR.
> >
> > * Berthelot, D., Raffel, C., Roy, A., & Goodfellow, I. (2018, September). Understanding and Improving Interpolation in Autoencoders via an Adversarial Regularizer. In International Conference on Learning Representations
> >
> > * Connor, M., & Rozell, C. (2020, April). Representing closed transformation paths in encoded network latent space. In Proceedings of the AAAI Conference on Artificial Intelligence (Vol. 34, No. 04, pp. 3666-3675).
> >
> > Generated paths are perceptually smooth so of course it means there is a correlation between latent space distances and perception.
> >
> > I also maintain that the submitted paper despite proposing a pluridisciplinary approach (Deep Learning based texture synthesis, psychophysics, neurosciences) does not provide any contribution in neurosciences or psychophysics.
> > Here are few examples of relevant pluridisciplinary papers that are not cited. They all use statistical representation of textures (can also be called a latent space) and study the perception of these textures using manipulation in the latent space.
> >
> > * Freeman, J., Ziemba, C. M., Heeger, D. J., Simoncelli, E. P., & Movshon, J. A. (2013). A functional and perceptual signature of the second visual area in primates. Nature neuroscience, 16(7), 974-981.
> >
> > * Okazawa, G., Tajima, S., & Komatsu, H. (2015). Image statistics underlying natural texture selectivity of neurons in macaque V4. Proceedings of the National Academy of Sciences, 112(4), E351-E360.
> >
> > * Vacher, J., Davila, A., Kohn, A., & Coen-Cagli, R. (2020). Texture Interpolation for Probing Visual Perception. Advances in Neural Information Processing Systems, 33.
> >
> > Finally, I found other references using perceptual similarity metric to generate images:
> >
> > * Dosovitskiy, A., & Brox, T. (2016). Generating images with perceptual similarity metrics based on deep networks. Advances in neural information processing systems, 29, 658-666.
> >
> > * Snell, J., Ridgeway, K., Liao, R., Roads, B. D., Mozer, M. C., & Zemel, R. S. (2017, September). Learning to generate images with perceptual similarity metrics. In 2017 IEEE International Conference on Image Processing (ICIP) (pp. 4277-4281). IEEE.
> >
> >
> > My feeling is still mixed about this paper. Though I agree that ICLR is a good venue to discuss this type of topics and increasing their visibility goes in the right direction. I therefore increase my score to 6.

---

> > > ### Author Response · Authors · 2021-11-22
> > > **Paper revision**
> > >
> > > Many thanks for the thorough response. In this paper we go beyond the existing literature by giving steps towards the understanding of the connection between the three related fields -- not just two. We are providing a first numerical quantification of the relation between statistics and perception by empirically showing the correlation (Sec. 2) and explain why this relation between autoencoders and statistics can be translated into perception. As pointed out by the reviewer, most authors in the literature agree that this relation exist, and probably smooth internal representations are evidence for that, but how this relationship can be modelled remains an open research question. What we are offering here is a first step in designing possible mathematical models for these relations (Sec 3 and 4). We also show examples of how these relations can affect the learning procedure in practise through the double counting effect (Section 5).
> > >
> > > While the connection between texture synthesis, neural style transfer and the different amounts of selectivity for textures in the different stages of the human visual system is directly related with the issues raised in the manuscript, we do not enter this discussion. Note that, for the case of deep learning models, we restrict ourselves to representations within autoencoders (which would be the content in style transfer or the last stage in the human visual system). We agree that it is a related (important) problem, but this could be a work by itself.
> > >
> > > We really thank the reviewer for the references. While we were aware of several of them, some offer really interesting connections with related problems. Based on them, we have added a discussion on how the internal representation of machine learning models can be used to interpolate between images and textures.

---

### Official Review · Reviewer_2c14 · 2021-11-02

**Correctness:** 3
**Technical Novelty And Significance:** 3
**Empirical Novelty And Significance:** 3
**Recommendation:** 6
**Confidence:** 4

**Main Review:**

Strengths:
The paper was nice to read. Overall there are six (6) observations and 3 Eqns in main body of the paper, which formed the core part of the work, while 13 more in Appendices were supporting as proofs and extra information.
Specific contributions are:
(i) Pixel-CNN++ based image likelihoods are correlated with human psycho-physical distances;
(ii) Distances induced by AEs are correlated with Prob(Trng data), as well as with human perception.
(ii) double counting effect in perceptual distance based loss function
Overall, as nice 3-way relationship being empirically explored.


Cons:
Let me start with some generic queries first:
I would like a comment from authors- these 6 observations are like soft theories (ala Perception theories),
or they can be raised to the level of Results/Lemmas (in applied maths )?
What can be the overall conclusion for theorists and experimental application domain experts, from these 6 (+ 3) contributions ?
How can one exploit them for better design of DL architectures/algos. to solve certain problems ?
I am trying to get an overall picture, if possible, of what is the gain in technical know-how or analytics this paper has produced.

The words Perceptual and psycho-physical distances - are they meaning the same ? A reference would be ideal.

You have used AE as your target application, what about other deep-CNN models - transformers, LSTM, GAN based models for generation or discrimination?

What does one need to do to counter this double-counting effect ? Modify loss function, dataset samples or  architecture ?
It appears that you have identified a tricky fallacy, but the remedy has not been explored ? Or, have I missed something.
You mention using training without image data - using uniform random noise as input- is this not same as various GAN-based models ?

Eqn (2) appears to be a special/modified form of Vapnik's theory of empirical risk minimization (ERM), applied to VAE ? What is the value addition then?

Eqn (3) - Ex is not defined ? Or I missed - although appears to be the trivial  - Empirical mean ? (I assume).

Any particular reason of Using  softplus (yes, recently popular) rather than softmax ? Pixel-CNN+ also uses softmax.

sec. 5.2, 1st para, pp 7 - this part of the (long) sentence may need rewording:
......minimizing a loss weighted by the "likelihood the samples" belong to the Student-t,....

Same within Sec. 6, 1st para, pp 8:
......by construction, for perceptual distances we are , considering that the human visual system has....
Now, this evolution process has been over a very long time, say more than few centuries  - and with no empirical results feasible/available involving ancestors of humans - my opinion is that:
 - this hypothesis used by visual perceptions scientists may be kept aside, in the context of this paper.

The last sentence in Conclusion section, pp 9, says:
....both machine learning and biological perception is informed by the distribution of natural images.

Well as scientists and engineers, should we not be more interested in the gap between the two, rather than the commonality ?
That would help us to bridge the gap and build more efficient, robust, autonomous, intelligent perception-machines, I believe,
as our main target to design.


Well the term "Appendices" is missing as a sub-heading before Appendix A - left me initially wondering if these were supplementary materials or Appendices of the paper. or extension.
I am not happy with the purple background used in figures 11-13. Quite difficult to isolate out the central part (hoping that is significant?)  visually - eye-stressing I must admit.
Figs. 18-19: traditional Image quality experts provide a zoomed-in part of pics to highlight the distortions. The overall pictures here look bland, and quite difficult to identify regions/parts of images with good vs bad quality (as evaluated often using SSIM or PSNR) - mostly, the pair in last row,
This is not the case for samples in Fig. 16.

Sec. E- Entropy limited AE: is this your idea, or a variant of any used elsewhere  ?
Did not see a reference - hence checking.

I enjoyed reading the paper, which had a nice flow, although a bit long (as per expectations in a Conf.)
due to supplementary/Appendices.



**Summary Of The Paper:**


The work presented in this paper aims to analyze the relationships between the probability distribution of the data, perceptual
distances, and unsupervised machine learning. Perceptual sensitivity is correlated with the probability of an image
in its close neighborhood. The paper also explores the relation between distances induced by autoencoders and the
probability distribution of the training data, as well as how these induced distances
are correlated with human perception. At the end, the paper specifies that perceptual distances do not
always lead to noticeable gains in performance over Euclidean distance in common
image processing tasks.

**Summary Of The Review:**


The paper has provided sufficient and substantial evidence of the observations made of the 3-way connections between -
learned or hand-crafted image representations (autoencoders and perceptual models), the distribution of
natural images, and human perception.

This paper may initiate good discussions/work along the domains overlapping visual perception and ML/DL scientists .
In terms of technical know-how, although the paper takes a small step, but does not really contribute to a big
gain in analytics and new formulations/algorithms.

Although there are some unanswered questions and issues, the paper appears quite interesting.
Minor improvements would be necessary for it to get accepted.

---

> ### Author Response · Authors · 2021-11-16
> **Response to Reviewer 2c14**
>
> Thank you for your comments!. The reviewer is right: observations in the paper are meant to play the role of soft theories rather than analytic results, however we think it is important to highlight these points to raise awareness and start a debate of why these (sometimes counter intuitive) phenomena happen in practice. In fact, we feel that ICLR could be an excellent venue to have this debate between scientists from the communities interested in *representations* (both machine learning and visual neuroscience) so that this link can be further explored in the future.
>
> Regarding practical consequences, the overall conclusions for domain experts are outlined in the final remarks, namely, that people should be aware of the effect of using perceptual distances to train machine learning models (whether they have a net positive or negative effect). We believe that practitioners need to take these consequences into account when designing models / loss functions in image tasks. We have not suggested a set of ‘best practices’ because this entirely depends on the aim of the practitioner. As a first approximation one could counteract the double-counting effect by weighting by the inverse of the probability (if one has a trustable probability model for the dataset at hand). However, it is not totally clear whether the practitioner would want to do so as, as commented in the manuscript, this double-counting effect would enforce a regularization which can be beneficial in situations with low data (for instance, Figure 20).
>
> We thought we had highlighted the implications of the observations (actually we included two experiments to show these in Figs. 5, 18, 19 and 20). However, we have devoted additional text on page 9 to reinforce the message.
>
> With regards to alternative models, perceptual measures like LPIPS exploit classification networks to create a perceptual metric, suggesting that at least some of the behaviour generalises to other deep CNN models. However, we think this needs more exploration. Training GAN-based models is different from training using random noise as the input - with GANs, the discriminator is presented with real images in training, whereas in our example at no time is the model presented with real images. The structure of images is purely learnt from the loss function being a perceptual metric.
>
> > Eqn (2) appears to be a special/modified form of Vapnik's theory of empirical risk minimization (ERM), applied to VAE ? What is the value addition then?
>
> Equation 2 is a form of ERM and we do not propose this to be a novel contribution, but rather show the reader that the training distribution is in the equation for the true risk explicitly when integrating over all space $\mathbf{x}$.
>
> >Eqn (3) - Ex is not defined ? Or I missed - although appears to be the trivial - Empirical mean ? (I assume).
>
> We edited the paper to define $\mathbb{E}_x$. Besides we have cited the V-C theory here as a more general work than the one cited.
>
> > Any particular reason of Using softplus (yes, recently popular) rather than softmax ? Pixel-CNN+ also uses softmax.
>
> We use the softplus activation as we did not wish to suggest our own architecture. This could bias the results and therefore we wanted to use pre-existing architectures. This particular architecture is taken from [1].
>
> > Well as scientists and engineers, should we not be more interested in the gap between the two, rather than the commonality ? That would help us to bridge the gap and build more efficient, robust, autonomous, intelligent perception-machines, I believe, as our main target to design.
>
> Whilst designing better models is the end goal, we believe one must first understand the effects of the different factors at play. This paper presents a number of observations displaying the effect of training machine learning models using perceptual metrics which we believe to be fundamental for practitioners. The overarching aim of the paper is to start a discussion with the community that eventually may lead to designing more efficient models.
>
> > Sec. E- Entropy limited AE: is this your idea, or a variant of any used elsewhere ? Did not see a reference - hence checking.
>
> The entropy limited autoencoder is not our idea, it was proposed in [2] and used as a way of comparing perceptual metrics in [3], we have added the references.
>
> [1] Ballé, Johannes, et al. "Nonlinear transform coding." IEEE Journal of Selected Topics in Signal Processing 15.2 (2020): 339-353.
>
> [2] Agustsson, Eirikur, et al. "Generative adversarial networks for extreme learned image compression." Proceedings of the IEEE/CVF International Conference on Computer Vision. 2019.
>
> [3] Ding, Keyan, et al. "Comparison of full-reference image quality models for optimization of image processing systems." International Journal of Computer Vision 129.4 (2021): 1258-1281.

---

> ### Author Response · Authors · 2021-11-29
> **Response 2 to Reviewer 2c14**
>
> Please let us know your thoughts regarding our comments in the previous post. We'd love to know your opinion and if you are happy with our responses and changes made to the paper, please feel free to update your score.

---

### Official Review · Reviewer_z8xf · 2021-11-05

**Correctness:** 2
**Technical Novelty And Significance:** 2
**Empirical Novelty And Significance:** 2
**Recommendation:** 6
**Confidence:** 5

**Main Review:**

Strengths:

The authors investigate a very interesting and important problem that lies in the intersection between perception and statistics.

Weaknesses:

For Observation 1:  $\frac{D_p(\mathbf{x}_1, \mathbf{x}_2)}{\Vert \mathbf{x}_2 - \mathbf{x}_1\Vert_2}$ is symmetric. Therefore, it should be correlated with both $p(\mathbf{x}_1)$ and $p(\mathbf{x}_2)$. One subtlety is one image (e.g., $\mathbf{x}_2$) is a "distorted" version of the other (e.g., $\mathbf{x}_1$). In other words, $p(\mathbf{x}_1)$ is the distribution over undistorted data and $p(\mathbf{x}_2)$ is the distribution over distorted data. Does it make sense that $\frac{D_p(\mathbf{x}_1, \mathbf{x}_2)}{\Vert \mathbf{x}_2 - \mathbf{x}_1\Vert_2}$ is correlated with two different distributions. Note that although Observation 1 assumes $\Vert \mathbf{x}_2 - \mathbf{x}_1\Vert_2$ is small, $p(\mathbf{x}_2)$ can be a distribution over very perceptually noisy data.


For Observation 2: This observation relies on the idea of score matching, which is derived under a particular type of distortion -  additive white Gaussian noise. Does it generalize to other types of non-additive, non-linear distortions, e.g., JPEG compression, Gaussian blurring, etc.

For Observation 3: The reviewer believes this observation will largely depend on the capacity (i.e., information bottleneck) of $f$. One extreme counterexample is that $f$ has no capacity and is a constant function.

For Observation 4: $p(\mathbf{x}) = p(e(\mathbf{x}))\left\vert\frac{\partial e}{\partial \mathbf{x}}(\mathbf{x})\right\vert$ holds only for bijective mappings with identical input and output dimensions. This is not the case for compression. In addition, comments on Objective 3 can also be applied here.

For Observation 5: The reviewer respectively disagrees with this observation. For $D_\mathrm{in}(\mathbf{x}_1,\mathbf{x}_2) = \Vert e(\mathbf{x}_1) - e(\mathbf{x}_2)\Vert_2$, it is not hard to find, for example using gradient-based optimization, a distorted image $\mathbf{x}_2$ with the same compressed representation as the original image, $\mathbf{x}_1$ of perfect quality, that is, $e(\mathbf{x}_1) = e(\mathbf{x}_2)$. This image may not appear in TID 2013, on which the authors justify Observtation 5.

Moreover, it remains to be seen whether PixelCNN++ is a good generative model to approximate $p(\mathbf{x})$, and how the approximation error affects the five observations.

A final concern is that there is a lack of experiments/discussions on how useful these observations are for designing better generative models and/or better perceptual distances.




**Summary Of The Paper:**

Interesting (but debatable) observations between image prior, image quality, and perceptual distances.

**Summary Of The Review:**

See above for detailed comments.

---

> ### Author Response · Authors · 2021-11-16
> **Response to Reviewer z8xf**
>
> Thank you for your review. We address each of the comments below.
>
> >For Observation 1: $\frac{D_p(\mathbf{x_1},\mathbf{x_2})}{\textrm{RMSE}(\mathbf{x_1},\mathbf{x_2})}$ is symmetric. Therefore, it should be correlated with both $p(\mathbf{x_1})$  and  $p(\mathbf{x_2})$. One subtlety is one image (e.g., $\mathbf{x_2}$) is a "distorted" version of the other (e.g., $\mathbf{x_1}$). In other words, $p(\mathbf{x_1})$  is the distribution over undistorted data and $p(\mathbf{x_2})$ is the distribution over distorted data. Does it make sense that $\frac{D_p(\mathbf{x_1},\mathbf{x_2})}{\textrm{RMSE}(\mathbf{x_1},\mathbf{x_2})}$ is correlated with two different distributions. Note that although Observation 1 assumes $||\mathbf{x_1} - \mathbf{x_2}||_2$ is small,  can be a distribution over very perceptually noisy data.
>
> $p(\mathbf{x})$ is the probability of image $\mathbf{x}$ belonging to the distribution of natural images $p$. There is no distribution over the distorted data -- rather, the distorted data should lie in low probability regions of $p$ and therefore $\frac{D_p(\mathbf{x_1},\mathbf{x_2})}{\textrm{RMSE}(\mathbf{x_1},\mathbf{x_2})}$ is only correlated with the probability of the original image, $p(\mathbf{x_1})$. We have highlighted this in the new version.
> Whilst Observation 1 does assume a small $||\mathbf{x_1} - \mathbf{x_2}||_2$, we have tested with large amounts of noise and still find a clear correlation.
>
> >For Observation 2: This observation relies on the idea of score matching, which is derived under a particular type of distortion - additive white Gaussian noise. Does it generalize to other types of non-additive, non-linear distortions, e.g., JPEG compression, Gaussian blurring, etc.
>
> Observation 2 does not rely on the idea of score matching, but it is along a similar vein. Observation 2 concerns both the image $\mathbf{x}$ and the reconstruction of the image $\mathbf{\hat{x}}$, so any distortion is due to the reconstruction error in the autoencoder. Whilst score matching says that this distance is related to the derivative of the log probability under additive Gaussian noise, we simply say that any reconstruction error from an autoencoder is correlated with the inverse of the probability. Score matching has been proven to be more general than Gaussian noise [1], and our observation (based on Eq. 4) is even more general than this because Eq. 4 is not attached to any particular noise/metric. Our observation does not have the same analytical expression as the score matching result, but in Appendix B we see our observation is consistent with the score matching solution around the mode of the PDF.
>
> > For Observation 3: The reviewer believes this observation will largely depend on the capacity (i.e., information bottleneck) of $f$. One extreme counterexample is that $f$ has no capacity and is a constant function.
>
> It is true that if the entropy is very restricted, $f$ has no capacity and in the limit may even be a constant function - however this is a very extreme case and, in practice, an autoencoder like this would never be used. We tested a range of reasonable rates (Fig. 2) and found that the correlation does depend on the restriction on the entropy - but even in the opposite extreme of a high rate regime, we still observe a positive correlation.
>
> > For Observation 4:  $p(\mathbf{x}) = p(e(\mathbf{x}))|\frac{\partial e}{\partial \mathbf{x}}(\mathbf{x})|$ holds only for bijective mappings with identical input and output dimensions. This is not the case for compression. In addition, comments on Objective 3 can also be applied here.
>
> The relationship that inspired Observation 4, $p(\mathbf{x}) = p(e(\mathbf{x}))|\frac{\partial e}{\partial \mathbf{x}}(\mathbf{x})|$, does rely on the transformation not altering the dimensions, however, in cases where dimension is not preserved one can embed the input and output into spaces of larger (and equal) dimensions and add negligible noise to avoid flatness in the manifold. In that equivalent case, the referred equation would hold.
> Regarding the other comment on a constant $f$, as earlier, the extreme example of $f$ having no capacity is an extreme example, and in practice, no autoencoder would work like that.

---

> ### Author Response · Authors · 2021-11-16
> **Response to Reviewer z8xf Part 2**
>
> > For Observation 5: The reviewer respectively disagrees with this observation. For $D_in(\mathbf{x_1}, \mathbf{x_2})=||e(\mathbf{x_1}) - e(\mathbf{x_2})||_2$, it is not hard to find, for example using gradient-based optimization, a distorted image $\mathbf{x_2}$ with the same compressed representation as the original image, $\mathbf{x_1}$ of perfect quality, that is, $e(\mathbf{x_1}) = e(\mathbf{x_2})$. This image may not appear in TID 2013, on which the authors justify Observtation 5.
>
> The case of two signals that have a non zero Euclidean distance but 0 distance in the embedded domain is similar to metamers in humans - physically different objects that elicit the same internal representation and thus have 0 perceptual distance (are perceived as the same). It makes sense that these exist, but the existence of data points not included in the considered dataset where the observation is tested does not reduce the validity of the experiment. In fact (thanks for the suggestion!) it would be very interesting to generate such samples and actually have them rated by humans in order to include them in subjective distortion databases. It may happen that the subjective distortion reported by humans for these samples is very low (which would confirm our point). However, that is an experimental issue that is out of the scope here.
>
> As a general comment, we give empirical evidence of the observations and offer plausible explanations and discussion of these observations. This is why we decided to refer to them as observations rather than hypotheses. We totally agree that the cause, consequences, and applicability of the observations are arguably - in fact this is the main idea of the manuscript: to raise awareness and start a debate of why these (sometimes counter intuitive) phenomena happen in practice.
>
> > Moreover, it remains to be seen whether PixelCNN++ is a good generative model to approximate $p(\mathbf{x})$, and how the approximation error affects the five observations.
>
> We use PixelCNN++ as an approximation to $p(\mathbf{x})$, as it achieves good negative log-likelihoods on the dataset we are evaluating it on, CIFAR-10. We did consider several different probability models but PixelCNN++ is arguably the best fit for our setting. However if the reviewer knows of any other good approximates for $p(\mathbf{x})$, we would appreciate being made aware of them.
>
> Whilst we have no experiments regarding designing better generative models, this paper is meant as an initial work in the field, linking the three topics; perception, image probability and machine learning models. We hope to begin a discussion regarding the observations in the paper that should lead to designing better image probability models. We showed 2 examples of an application of the behaviour we observed (one in the main text and another in the appendix) but we rewrote some discussion of possible applications to talk about using them in the design of generative models in our conclusions
>
> [1] Alain, Guillaume, and Yoshua Bengio. "What regularized auto-encoders learn from the data-generating distribution." The Journal of Machine Learning Research 15.1 (2014): 3563-3593.

---

> > ### Comment · Reviewer_z8xf · 2021-11-17
> > **Comments Well Addressed**
> >
> > Thank the authors for the point-for-point responses, which address most of the reviewer's concerns. The reviewer agrees with the authors in that the main contribution of this work is making links among three important topics that are highly relevant to ICLR, perception, image probability, and practical machine learning by drawing the five observations with empirical justifications. Thus, I vote for the acceptance of this manuscript with one minor comment:
> >
> > The authors may give a more accurate description of the five observations based on the reviewer's previous comments. E.g., discuss circumstances, on which these observations are not likely to be true in practice.

---

> > > ### Author Response · Authors · 2021-11-21
> > > **Rebuttal Revision**
> > >
> > > Thank you for your comments. We have included some comments in the paper on if $f$ has no capacity and the existence of metamers in the human visual system and in our models.

---

### Decision · Program_Chairs · 2022-01-20

**Decision:**

Accept (Spotlight)

**Comment:**

All reviewers suggest acceptance of this paper, which reports the relationship between perceptual distances, data distributions, and contemporary unsupervised machine learning methods.  I believe this paper will be of broad interest to different communities at ICLR.